# Trajectory of Alternating Direction Method of Multipliers and Adaptive Acceleration

**Clarice Poon**[*]
University of Bath, Bath UK
cmhsp20@bath.ac.uk

**Jingwei Liang**[*]
University of Cambridge, Cambridge UK
jl993@cam.ac.uk

## Abstract

The alternating direction method of multipliers (ADMM) is one of the most widely used first-order methods in the literature owing to its simplicity, flexibility and efficiency. Over the years, numerous efforts are made to improve the performance of ADMM, such as the inertial technique. By studying the geometric properties of ADMM, we discuss the limitations of current inertial accelerated ADMM, then present and analyze an adaptive acceleration scheme for the method. Numerical experiments on problems arising from image processing, statistics and machine learning demonstrate the advantages of the proposed acceleration approach.

## 1 Introduction

Consider the following constrained and composite optimisation problem

$$\min_{x \in \mathbb{R}^n, y \in \mathbb{R}^m} R(x) + J(y) \quad \text{such that} \quad Ax + By = b, \qquad (\mathcal{P}_{\text{ADMM}})$$

where the following basic assumptions are imposed

($\boldsymbol{\mathcal{A}}$.1) $R \in \Gamma_0(\mathbb{R}^n)$ and $J \in \Gamma_0(\mathbb{R}^m)$ are proper convex and lower semi-continuous functions.

($\boldsymbol{\mathcal{A}}$.2) $A : \mathbb{R}^n \to \mathbb{R}^p$ and $B : \mathbb{R}^m \to \mathbb{R}^p$ are injective linear operators.

($\boldsymbol{\mathcal{A}}$.3) $\text{ri}(\text{dom}(R) \cap \text{dom}(J)) \neq \emptyset$, and the set of minimizers is non-empty.

Over the past years, problem ($\mathcal{P}_{\text{ADMM}}$) has attracted a great deal of interests as it covers many problems arising from data science, machine learning, statistics, inverse problems and imaging, etc.; See Section 5 for examples. In the literature, different methods are proposed to handle the problem, among them the alternating direction method of multipliers (ADMM) is the most prevailing one.

Earlier works of ADMM include [17, 16, 15, 11], and recently it has gained increasing popularity, in part due to [6]. To derive ADMM, first consider the augmented Lagrangian associated to ($\mathcal{P}_{\text{ADMM}}$) $\mathcal{L}(x, y; \psi) \stackrel{\text{def}}{=} R(x) + J(y) + \langle \psi, Ax + By - b \rangle + \frac{\gamma}{2} \|Ax + By - b\|^2$, where $\gamma > 0$ and $\psi \in \mathbb{R}^p$ is the Lagrangian multiplier. To find a saddle-point of $\mathcal{L}(x, y; \psi)$, ADMM applies the iteration

$$
\begin{aligned}
x_k &= \text{argmin}_{x \in \mathbb{R}^n} R(x) + \tfrac{\gamma}{2} \|Ax + By_{k-1} - b + \tfrac{1}{\gamma} \psi_{k-1}\|^2, \\
y_k &= \text{argmin}_{y \in \mathbb{R}^m} J(y) + \tfrac{\gamma}{2} \|Ax_k + By - b + \tfrac{1}{\gamma} \psi_{k-1}\|^2, \\
\psi_k &= \psi_{k-1} + \gamma(Ax_k + By_k - b).
\end{aligned}
\qquad (1)
$$

By defining $z_k \stackrel{\text{def}}{=} \psi_{k-1} + \gamma Ax_k$, we can rewrite ADMM iteration (1) into the following form

$$
\begin{aligned}
x_k &= \text{argmin}_{x \in \mathbb{R}^n} R(x) + \tfrac{\gamma}{2} \|Ax - \tfrac{1}{\gamma}(z_{k-1} - 2\psi_{k-1})\|^2, \\
z_k &= \psi_{k-1} + \gamma Ax_k, \\
y_k &= \text{argmin}_{y \in \mathbb{R}^m} J(y) + \tfrac{\gamma}{2} \|By + \tfrac{1}{\gamma}(z_k - \gamma b)\|^2, \\
\psi_k &= z_k + \gamma(By_k - b).
\end{aligned}
\qquad (2)
$$

For the rest of the paper, we will consider the above four-point formulation.

---

[*]Equal contributions.

**Contributions** The contribution of our paper is threefold. First, for the sequence $\{z_k\}_{k\in\mathbb{N}}$ of (2), we show that it has two different types of trajectory:

- When both $R, J$ are non-smooth functions, under the assumption that they are partly smooth (see Definition 2.1), we show that the eventual trajectory of $\{z_k\}_{k\in\mathbb{N}}$ is approximately a spiral which can be characterized precisely if $R, J$ are moreover locally polyhedral around the solution.
- When at least one of $R, J$ is smooth, we show that depends on the choice of $\gamma$, the eventual trajectory of $\{z_k\}_{k\in\mathbb{N}}$ can be either straight line or spiral.

Second, based on trajectory of $\{z_k\}_{k\in\mathbb{N}}$, we discuss the limitations of the current combination of ADMM and inertial acceleration technique. In Section 3, we distinguish the situations where inertial acceleration will work and when it fails. More precisely: inertial technique will work if the trajectory of $\{z_k\}_{k\in\mathbb{N}}$ is or close to a straight line, and will fail if the trajectory is a spiral.

Our core contribution is an adaptive acceleration for ADMM, which is inspired by the trajectory of ADMM and dubbed "A$^3$DMM". The limitation of inertial technique, particularly its failure, implies that the right acceleration scheme should be able to follow the trajectory of the iterates. In Section 4, we propose an adaptive linear prediction scheme to accelerate ADMM which is able to following the trajectory of the method. Our proposed A$^3$DMM belongs to the realm of extrapolation method, and provides an alternative geometrical interpretation for polynomial extrapolation methods such as Minimal Polynomial Extrapolation (MPE) [9] and Reduced Rank Extrapolation (RRE) [12, 21].

**Related works** Over the past decades, owing to the tremendous success of inertial acceleration [22, 5], the inertial technique has been widely adapted to accelerate other first-order methods. In terms of ADMM, related work can be found in [23, 18, 14], either from proximal point algorithm perspective or continuous dynamical system. However, to ensure that inertial acceleration works, strong assumptions are imposed on $R, J$ in ($\mathcal{P}_{\text{ADMM}}$), such as smooth differentiability or strong convexity. When it comes to general non-smooth problems, these works may fail to provide acceleration. Recently in [13], an $O(1/k^2)$ convergence rate is established for ADMM using Nesterov acceleration, however the result holds only for the continuous dynamical system while the discrete-time optimization scheme remains unavailable.

For more generic acceleration techniques, there are extensive works in numerical analysis on the topic of convergence acceleration for sequences. The goal of convergence acceleration is, given an arbitrary sequence $\{z_k\}_{k\in\mathbb{N}} \subset \mathbb{R}^n$ with limit $z^\star$, finding a transformation $\mathcal{E}_k : \{z_{k-j}\}_{j=1}^q \to \bar{z}_k \in \mathbb{R}^n$ such that $\bar{z}_k$ converges faster to $z^\star$. In general, the process by which $\{z_k\}_{k\in\mathbb{N}}$ is generated is unknown, $q$ is chosen to be a small integer, and $\bar{z}_k$ is referred to as the extrapolation of $z_k$. Some of the best known examples include Richardson's extrapolation [24], the $\Delta^2$-process of Aitken [1] and Shank's algorithm [26]. We refer to [7, 8, 27] and references therein for a detailed historical perspective on the development of these techniques. Much of the works on the extrapolation of vector sequences was initiated by Wynn [29] who generalized the work of Shank to vector sequences. In the supplementary material, the formulation of some of these methods are provided. In particular, minimal polynomial extrapolation (MPE) [9] and Reduced Rank Extrapolation (RRE) [12, 21] (which is also a variant of Anderson acceleration developed independently in [3]), which are particularly relevant to this present work (see Section 4.2 for brief discussion).

More recently, there has been a series of work on a regularised version of RRE stemming from [25]. We remark however the regularisation parameter in these works rely on a grid search based on objective function, their applicability to the general ADMM setting is unclear.

**Notations** Denote $\mathbb{R}^n$ a $n$-dimensional Euclidean space equipped with scalar product $\langle \cdot, \cdot \rangle$ and norm $\| \cdot \|$. Id denotes the identity operator on $\mathbb{R}^n$. $\Gamma_0(\mathbb{R}^n)$ denotes the class of proper convex and lower-semicontinuous functions on $\mathbb{R}^n$. For a nonempty convex set $S \subset \mathbb{R}^n$, denote $\text{ri}(S)$ its relative interior, $\text{par}(S)$ the smallest subspace parallel to $S$ and $\mathcal{P}_S$ the projection operator onto $S$. The sub-differential of a function $R \in \Gamma_0(\mathbb{R}^n)$ is defined by $\partial R(x) \stackrel{\text{def}}{=} \{g \in \mathbb{R}^n | R(x') \geq R(x) + \langle g, x' - x \rangle, \forall x' \in \mathbb{R}^n\}$. The spectral radius of a matrix $M$ is denoted by $\rho(M)$.

## 2 Trajectory of ADMM

In this section, we discuss the trajectory of the sequence $\{z_k\}_{k\in\mathbb{N}}$ generated by ADMM based on the concept "partial smoothness" which was first introduced in [19].

### 2.1 Partial smoothness

Let $\mathcal{M} \subset \mathbb{R}^n$ be a $C^2$-smooth submanifold, denote $\mathcal{T}_\mathcal{M}(x)$ the tangent space of $\mathcal{M}$ at a point $x \in \mathcal{M}$.

**Definition 2.1 (Partly smooth function [19]).** A function $R \in \Gamma_0(\mathbb{R}^n)$ is partly smooth at $\bar{x}$ relative to a set $\mathcal{M}_{\bar{x}}$ if $\partial R(\bar{x}) \neq \emptyset$ and $\mathcal{M}_{\bar{x}}$ is a $C^2$ manifold around $\bar{x}$, and moreover

    **Smoothness** $R$ restricted to $\mathcal{M}_{\bar{x}}$ is $C^2$ around $\bar{x}$.
    **Sharpness** The tangent space $\mathcal{T}_{\mathcal{M}_{\bar{x}}}(\bar{x}) = \mathrm{par}(\partial R(\bar{x}))^{\perp}$.
    **Continuity** The set-valued mapping $\partial R$ is continuous at $x$ relative to $\mathcal{M}_{\bar{x}}$.

The class of partly smooth functions at $\bar{x}$ relative to $\mathcal{M}_{\bar{x}}$ is denoted as $\mathrm{PSF}_{\bar{x}}(\mathcal{M}_{\bar{x}})$. Popular examples of partly smooth functions can be found in [20, Chapter 5]. Loosely speaking, a partly smooth function behaves *smoothly* as we move along $\mathcal{M}_{\bar{x}}$, and *sharply* if we move transversal to it.

## 2.2 Trajectory of ADMM

The iteration of ADMM is non-linear in general owing to the non-smoothness and non-linearity of $R$ and $J$. However, if they are partly smooth, the local $C^2$-smoothness allows us to linearize the ADMM iteration, and hence enables us to study the trajectory of sequence generated by the method. We denote $(x^{\star}, y^{\star}, \psi^{\star})$ a saddle-point of $\mathcal{L}(x, y; \psi)$ and let $z^{\star} = \psi^{\star} + \gamma A x^{\star}$.

To discuss the trajectory of ADMM, we rely on sequence $\{z_k\}_{k \in \mathbb{N}}$. Define $v_k \overset{\text{def}}{=} z_k - z_{k-1}$ and $\theta_k \overset{\text{def}}{=} \arccos(\frac{\langle v_k, v_{k-1} \rangle}{\|v_k\| \|v_{k-1}\|})$ the angle between $v_k, v_{k-1}$. We use $\{\theta_k\}_{k \in \mathbb{N}}$ to characterize the trajectory of $\{z_k\}_{k \in \mathbb{N}}$. Given $(x^{\star}, y^{\star}, \psi^{\star})$, the first-order optimality condition entails $-A^T \psi^{\star} \in \partial R(x^{\star})$ and $-B^T \psi^{\star} \in \partial J(y^{\star})$, below we impose

$$- A^T \psi^{\star} \in \mathrm{ri}\big(\partial R(x^{\star})\big) \quad \text{and} \quad - B^T \psi^{\star} \in \mathrm{ri}\big(\partial J(y^{\star})\big). \tag{ND}$$

**Both $R, J$ are non-smooth** Let $\mathcal{M}_{x^{\star}}^R, \mathcal{M}_{y^{\star}}^J$ be two smooth manifolds around $x^{\star}, y^{\star}$ respectively, and suppose $R \in \mathrm{PSF}_{x^{\star}}(\mathcal{M}_{x^{\star}}^R), J \in \mathrm{PSF}_{y^{\star}}(\mathcal{M}_{y^{\star}}^J)$ are partly smooth. Denote $T_{x^{\star}}^R, T_{y^{\star}}^J$ the tangent spaces of $\mathcal{M}_{x^{\star}}^R, \mathcal{M}_{y^{\star}}^J$ at $x^{\star}, y^{\star}$, respectively. Let $A_R \overset{\text{def}}{=} A \circ \mathcal{P}_{T_{x^{\star}}^R}, B_J \overset{\text{def}}{=} B \circ \mathcal{P}_{T_{y^{\star}}^J}$ and $T_{A_R}, T_{B_J}$ be the range of $A_R, B_J$ respectively. Denote $(\alpha_j)_{j=1,\ldots}$ the Principal angles (see Section D.2 in the supplementary for definition) between $T_{A_R}, T_{B_J}$, and let $\alpha_F, \alpha'$ be the smallest and 2nd smallest of $\alpha_j$ which are larger than 0.

**Theorem 2.2.** *For problem* ($\mathcal{P}_{\mathrm{ADMM}}$) *and ADMM iteration* (1)*, assume that conditions* (<span style="color:red">A</span>.1)-(<span style="color:red">A</span>.3) *are true, then* $(x_k, y_k, \psi_k)$ *converges to a saddle point* $(x^{\star}, y^{\star}, \psi^{\star})$ *of* $\mathcal{L}(x, y; \psi)$*. Suppose that* $R \in \mathrm{PSF}_{x^{\star}}(\mathcal{M}_{x^{\star}}^R), J \in \mathrm{PSF}_{y^{\star}}(\mathcal{M}_{y^{\star}}^J)$ *and condition* (ND) *holds, then*

    (i) *There exists a matrix $M$ such that $v_k = M v_{k-1} + o(\|v_{k-1}\|)$ holds for all $k$ large enough.*
    (ii) *If moreover, $R, J$ are locally polyhedral around $x^{\star}, y^{\star}$, then $v_k = M v_{k-1}$ with $M$ being normal and having eigenvalues of the form $\cos(\alpha_j) e^{\pm i \alpha_j}$, and $\cos(\theta_k) = \cos(\alpha_F) + O(\eta^{2k})$ with $\eta = \cos(\alpha')/\cos(\alpha_F)$.*

**Remark 2.3.** The result indicates that, when both $R, J$ are locally polyhedral, the trajectory of $\{z_k\}_{k \in \mathbb{N}}$ is a spiral. For the case $R, J$ being general partly smooth function, though we cannot prove, numerical evidence shows that the trajectory of $\{z_k\}_{k \in \mathbb{N}}$ could be either straight line or also a spiral.

**$R$ or/and $J$ is smooth** Now we consider the case that at least one function out of $R, J$ is smooth. For simplicity, consider that $R$ is smooth and $J$ remains non-smooth.

**Proposition 2.4.** *For problem* ($\mathcal{P}_{\mathrm{ADMM}}$) *and ADMM iteration* (1)*, assume that conditions* (<span style="color:red">A</span>.1)-(<span style="color:red">A</span>.3) *are true, then* $(x_k, y_k, \psi_k)$ *converges to a saddle point* $(x^{\star}, y^{\star}, \psi^{\star})$ *of* $\mathcal{L}(x, y; \psi)$*. Suppose $R$ is locally $C^2$ around $x^{\star}$, $J \in \mathrm{PSF}_{y^{\star}}(\mathcal{M}_{y^{\star}}^J)$ is partly smooth and condition* (ND) *holds for $J$, then Theorem 2.2(i) holds for all $k$ large enough. If moreover, $A$ is full rank square matrix, then all the eigenvalues of $M$ are real for $\gamma > \|(A^T A)^{-\frac{1}{2}} \nabla^2 R(x^{\star}) (A^T A)^{-\frac{1}{2}}\|$.*

**Remark 2.5.** The spectrum of $M$ is real, numerical evidence shows that the eventual trajectory of $\{z_k\}_{k \in \mathbb{N}}$ is a straight line, which is different from the case where both functions are non-smooth. If $o(\|v_{k-1}\|)$ is vanishing fast enough, we can also prove that $\theta_k \to 0$.

It should be emphasized that the trajectory is determined by the property of the *leading eigenvalue* of $M$. Therefore, for $\gamma \leq \|(A^T A)^{-\frac{1}{2}} \nabla^2 R(x^{\star}) (A^T A)^{-\frac{1}{2}}\|$, though $M$ will have complex eigenvalues, the leading one is not necessarily to be complex. As a result, the trajectory of $\{z_k\}_{k \in \mathbb{N}}$ could be either spiral (complex leading eigenvalue) or straight line (real leading eigenvalue).

In Figure 1 (a) and (c), we present two examples of the trajectory of ADMM. Subfigure (a) shows a spiral trajectory in $\mathbb{R}^2$ which is obtained from solving a polyhedral problem, while subfigure (c) is an eventual straight line trajectory in $\mathbb{R}^3$.

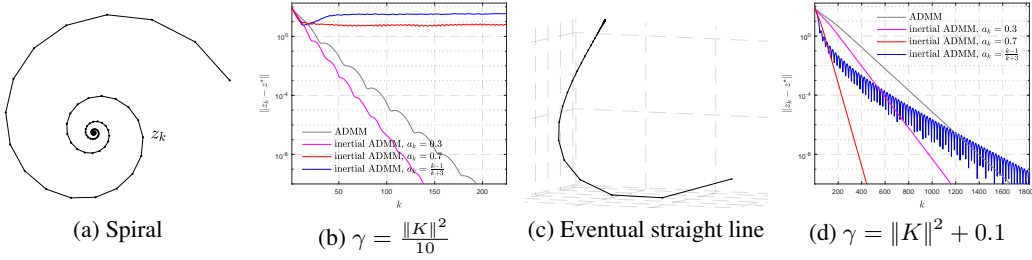

| (a) Spiral | (b) $\gamma = \frac{\|K\|^2}{10}$ | (c) Eventual straight line | (d) $\gamma = \|K\|^2 + 0.1$ |

Figure 1: Trajectory of sequence $\{z_k\}_{k\in\mathbb{N}}$ and effects of inertial on ADMM. (a) Spiral trajectory of ADMM; (b) failure of inertial ADMM on spiral trajectory; (c) Eventual straight line trajectory; (d) success of inertial ADMM on straight line trajectory.

## 3   The failure of inertial acceleration

One simple approach for combining inertial technique with ADMM is described below

$$
\begin{aligned}
x_k &= \mathrm{argmin}_{x\in\mathbb{R}^n}\, R(x) + \tfrac{\gamma}{2}\|Ax - \tfrac{1}{\gamma}(\bar{z}_{k-1} - 2\psi_{k-1})\|^2, \\
z_k &= \psi_{k-1} + \gamma A x_k, \\
\bar{z}_k &= z_k + a_k(z_k - z_{k-1}), \\
y_k &= \mathrm{argmin}_{y\in\mathbb{R}^m}\, J(y) + \tfrac{\gamma}{2}\|By + \tfrac{1}{\gamma}(\bar{z}_k - \gamma b)\|^2, \\
\psi_k &= \bar{z}_k + \gamma(B y_k - b),
\end{aligned}
\tag{3}
$$

which considers only the momentum of $\{z_k\}_{k\in\mathbb{N}}$ without any stronger assumptions on $R, J$. The above scheme can reformulated as an instance of inertial Proximal Point Algorithm, guaranteed to be convergent for $a_k < \tfrac{1}{3}$ [2]; We refer to [23] or [20, Chapter 4.3] for more details. To our knowledge, there is no acceleration guarantee for (3).

**Remark 3.1.** Besides (3), other combinations of inertial technique and ADMM are also proposed, see for instance [23, 18]. To ensure acceleration guarantees, stronger assumptions, such as Lipschitz smoothness and strong convexity, are needed.

We use LASSO problem to demonstrate the combination of the above inertial technique and ADMM, especially when it failures. The formulation of LASSO in the form of ($\mathcal{P}_{\mathrm{ADMM}}$) reads

$$
\min_{x,y\in\mathbb{R}^n} \mu\|x\|_1 + \tfrac{1}{2}\|Ky - f\|^2 \quad \text{such that} \quad x - y = 0,
\tag{4}
$$

where $K \in \mathbb{R}^{m\times n}$, $m < n$ is a random Gaussian matrix. Since $\tfrac{1}{2}\|Ky - f\|^2$ is quadratic, owing to Proposition 2.4, the eventual trajectory of $\{z_k\}_{k\in\mathbb{N}}$ is a straight line if $\gamma > \|K\|^2$, and a spiral for some $\gamma \leq \|K\|^2$. Therefore, we consider two different choices of $\gamma$ which are $\gamma = \|K\|^2/10$ and $\gamma = \|K\|^2 + 0.1$, and for each $\gamma$, four different choices of $a_k$ are considered

$$
a_k \equiv 0.3, \quad a_k \equiv 0.7 \quad \text{and} \quad a_k = \tfrac{k-1}{k+3}.
$$

The 3rd choice of $a_k$ corresponds to FISTA [10]. For the numerical example, we let $K \in \mathbb{R}^{640\times 2048}$ and $\mu = 1$, $f$ is the measurement of an 128-sparse signal. The results are shown in Figure 1 (b) & (d),

- $\gamma = \|K\|^2/10$: The inertial scheme works only for $a_k \equiv 0.3$, which is due to that fact that the trajectory of $\{z_k\}_{k\in\mathbb{N}}$ is a spiral for $\gamma = \|K\|^2/10$. As a result, the direction $z_k - z_{k-1}$ is not pointing towards $z^\star$, hence unable to provide satisfactory acceleration.
- $\gamma = \|K\|^2 + 0.1$: All choices of $a_k$ work since $\{z_k\}_{k\in\mathbb{N}}$ eventually forms a straight line. Among these four choices of $a_k$, $a_k \equiv 0.7$ is the fastest, while $a_k = \tfrac{k-1}{k+3}$ eventually is the slowest.

It should be noted that, though ADMM is faster under $\gamma = \|K\|^2/10$ than $\gamma = \|K\|^2 + 0.1$, our main focus here is to show how the trajectory of $\{z_k\}_{k\in\mathbb{N}}$ affects the outcome of inertial acceleration.

The above comparisons, particularly for $\gamma = \frac{\|K\|^2}{10}$, imply that the trajectory of $\{z_k\}_{k\in\mathbb{N}}$ is crucial for the acceleration outcome of the inertial scheme (3). Since the trajectory of $\{z_k\}_{k\in\mathbb{N}}$ depends on the properties of $R, J$ and choice of $\gamma$, this implies that the right scheme that can achieve uniform acceleration despite $R, J$ and $\gamma$ should be able to adapt itself to the trajectory of the method. More discussions on the failure of inertial can be found in Section A of the supplementary material.

# 4  A³DMM: adaptive acceleration for ADMM

The previous section shows the trajectory of $\{z_k\}_{k\in\mathbb{N}}$ eventually settles onto a regular path *i.e.* either straight line or spiral. In this section, we exploit this regularity to design adaptive acceleration for ADMM, which is called "A³DMM"; See Algorithm 1.

The update of $\bar{z}_k$ in (3) can be viewed as a special case of the following extrapolation

$$\bar{z}_k = \mathcal{E}(z_k, z_{k-1}, \cdots, z_{k-q-1}), \tag{5}$$

for the choice of $q = 0$. The idea is: given $\{z_{k-j}\}_{j=0}^{q+1}$, define $v_j \overset{\text{def}}{=} z_j - z_{j-1}$ and predict the future iterates by considering how the past directions $v_{k-1}, \ldots, v_{k-q}$ approximate the latest direction $v_k$. In particular, define $V_{k-1} \overset{\text{def}}{=} [v_{k-1}, \cdots, v_{k-q}] \in \mathbb{R}^{n\times q}$, and let $c_k \overset{\text{def}}{=} \operatorname{argmin}_{c\in\mathbb{R}^q} \|V_{k-1}c - v_k\|^2 = \|\sum_{j=1}^q c_j v_{k-j} - v_k\|^2$. The idea is then that $V_k c_k \approx v_{k+1}$ and so, $\bar{z}_{k,1} \overset{\text{def}}{=} z_k + V_k c \approx z_{k+1}$. By iterating this $s$ times, we obtain $\bar{z}_{k,s} \approx z_{k+s}$.

More precisely, given $c \in \mathbb{R}^q$, define the mapping $H$ by $H(c) = \begin{bmatrix} c_{1:q-1} & \mathrm{Id}_{q-1} \\ c_q & 0_{1,q-1} \end{bmatrix} \in \mathbb{R}^{q\times q}$. Let $C_k = H(c_k)$, note that $V_k = V_{k-1}C_k$. Define $\bar{V}_{k,0} \overset{\text{def}}{=} V_k$ and for $s \geq 1$, define $\bar{V}_{k,s} \overset{\text{def}}{=} \bar{V}_{k,s-1}C_k \overset{\text{def}}{=} V_k C_k^s$ where $C_k^s$ is the power of $C_k$. Let $(C)_{(:,1)}$ be the first column of matrix $C$, then

$$\bar{z}_{k,s} = z_k + \sum_{i=1}^s (\bar{V}_{k,i})_{(:,1)} = z_k + \sum_{i=1}^s V_k (C_k^i)_{(:,1)} = z_k + V_k \big(\sum_{i=1}^s C_k^i\big)_{(:,1)}, \tag{6}$$

which is the desired trajectory following extrapolation. Now define the extrapolation

$$\mathcal{E}_{s,q}(z_k, \cdots, z_{k-q-1}) \overset{\text{def}}{=} V_k \big(\sum_{i=1}^s C_k^i\big)_{(:,1)}$$

parameterized by $s, q$, we obtain the following trajectory following adaptive acceleration for ADMM.

---

**Algorithm 1:** A³DMM - Adaptive Acceleration for ADMM

---

**Initial**: Let $s \geq 1, q \geq 1$ be integers. Let $\bar{z}_0 = z_0 \in \mathbb{R}^p$ and $V_0 = 0 \in \mathbb{R}^{p\times(q+1)}$.
**Repeat**:
- For $k \geq 1$:
$$y_k = \operatorname{argmin}_{y\in\mathbb{R}^m} J(y) + \frac{\gamma}{2}\|By + \frac{1}{\gamma}(\bar{z}_{k-1} - \gamma b)\|^2,$$
$$\psi_k = \bar{z}_{k-1} + \gamma(By_k - b),$$
$$x_k = \operatorname{argmin}_{x\in\mathbb{R}^n} R(x) + \frac{\gamma}{2}\|Ax - \frac{1}{\gamma}(\bar{z}_{k-1} - 2\psi_k)\|^2,$$
$$z_k = \psi_k + \gamma Ax_k,$$
$$v_k = z_k - z_{k-1} \quad \text{and} \quad V_k = [v_k, V_{k-1}(:, 1:q-1)].$$
- If $\operatorname{mod}(k, q+2) = 0$: compute $c_k$ and $C_k$, if $\rho(C_k) < 1$: $\bar{z}_k = z_k + a_k\mathcal{E}_{s,q}(z_k, \cdots, z_{k-q-1})$.
**Until**: $\|v_k\| \leq \text{tol}$.

---

**Remark 4.1.**
- The extra computational cost of A³DMM is very small, which is about $nq^2$ for computing the pseudoinverse of $V_{k-1}$. And the value of $q$ usually is taken very small, *e.g.* $q \leq 10$.
- The reason we change the order of updates in Algorithm 1 is that the update of $y_k$ requires only $\bar{z}_k$, doing so we only need to extrapolate $z_k$ which requires the minimal computational overhead. Moreover, the extrapolation can also be applied to $x_k, y_k, \psi_k$ under proper adaptation.
- A³DMM carries out $(q + 2)$ standard ADMM iterations to set up the extrapolation step $\mathcal{E}_{s,q}$. As $\mathcal{E}_{s,q}$ contains the sum of the powers of $C_k$, it is guaranteed to be convergent when $\rho(C_k) < 1$. Therefore, we only apply $\mathcal{E}_{s,q}$ when the spectral radius $\rho(C_k) < 1$ is true. In this case, there is a closed form expression for $\mathcal{E}_{s,q}$ when $s = +\infty$; See Eq. (7).
- The purpose of adding $a_k$ in front of $\mathcal{E}_{s,q}(z_k, \cdots, z_{k-q-1})$ is so that we can control the value of $a_k$ to ensure the convergence of the algorithm; See below the discussion.

## 4.1  Convergence of A³DMM

To discuss the convergence of A³DMM, we shall treat the algorithm as a perturbation of the original ADMM. If the perturbation error is absolutely summable, then we obtain the convergence of A³DMM. More precisely, let $\varepsilon_k \in \mathbb{R}^n$ whose value takes

$$\varepsilon_k = \begin{cases} 0 : \operatorname{mod}(k, q+2) \neq 0 \ \text{ or } \ \operatorname{mod}(k, q+2) = 0 \ \& \ \rho(C_k) \geq 1, \\ a_k\mathcal{E}_{s,q}(z_k, \cdots, z_{k-q-1}) : \operatorname{mod}(k, q+2) = 0 \ \& \ \rho(C_k) < 1. \end{cases}$$

Suppose the fixed-point formulation of ADMM can be written as $z_k = \mathcal{F}(z_{k-1})$ for some $\mathcal{F}$ (see Section B.2 of the appendix for details). Then Algorithm 1 can be written as $z_k = \mathcal{F}(z_{k-1} + \varepsilon_{k-1})$, and we can obtain the following convergence for Algorithm 1 which is based on the classic convergence result of inexact Krasnosel'skiĭ-Mann fixed-point iteration [4, Proposition 5.34].

**Proposition 4.2.** *For problem* $(\mathcal{P}_{\mathrm{ADMM}})$ *and Algorithm 1, suppose that the conditions* (**A.1**)-(**A.3**) *are true. If moreover,* $\sum_k \|\varepsilon_k\| < +\infty$, $z_k \to z^\star \in \mathrm{fix}(\mathcal{F}) \overset{\text{def}}{=} \{z \in \mathbb{R}^p : z = \mathcal{F}(z)\}$ *and* $(x_k, y_k, \psi_k)$ *converges to* $(x^\star, y^\star, \psi^\star)$ *which is a saddle point of* $\mathcal{L}(x, y; \psi)$.

**On-line updating rule** The summability condition $\sum_k \|\varepsilon_k\| < +\infty$ in general cannot be guaranteed. However, it can be enforced by a simple online updating rule. Let $a \in [0,1]$ and $b, \delta > 0$, then $a_k$ can be determined by $a_k = \min\{a, b/(k^{1+\delta}\|z_k - z_{k-1}\|)\}$.

**Inexact A³DMM** Observe that in A³DMM, when $A, B$ are non-trivial, in general there are no closed form solutions for $x_k$ and $y_k$. Take $x_k$ for example, suppose it is computed approximately, then in $z_k$ there will be another approximation error $\varepsilon'_k$, and consequently

$$z_k = \mathcal{F}(z_{k-1} + \varepsilon_{k-1} + \gamma\varepsilon'_{k-1}).$$

If there holds $\sum_k \|\varepsilon'_{k-1}\| < +\infty$, Proposition 4.2 remains true for the above perturbation form.

## 4.2 Acceleration guarantee for A³DMM

We have so far alluded to the idea that the extrapolated point $\bar{z}_{k,s}$ defined in (6) (which depends only on $\{z_{k-j}\}_{j=0}^q$) is an approximation to $z_{k+s}$. In this section, we make precise this statement.

**Relationship to MPE and RRE** We first show that $\bar{z}_{k,\infty}$ is (almost) equivalent to MPE. Recall that given a square matrix $C$, if its Neumann series is convergent, then there holds $(\mathrm{Id}-C)^{-1} = \sum_{i=0}^{+\infty} C^i$. Now for the summation of the power of $C_k$ in (6), when $s = +\infty$, we have

$$\textstyle\sum_{i=1}^{+\infty} C_k^i = C_k \sum_{i=0}^{+\infty} C_k^i = C_k(\mathrm{Id}-C_k)^{-1} = (\mathrm{Id}-C_k)^{-1} - \mathrm{Id}.$$

Back to (6), then we get

$$
\begin{aligned}
\bar{z}_{k,\infty} &\overset{\text{def}}{=} z_k + V_k\big((\mathrm{Id}-C_k)^{-1} - \mathrm{Id}\big)_{(:,1)} = z_k - v_k + V_k\big((\mathrm{Id}-C_k)^{-1}\big)_{(:,1)} \\
&= z_{k-1} + V_k\big((\mathrm{Id}-C_k)^{-1}\big)_{(:,1)} = \tfrac{1}{1-\sum_{i=1}^s c_{k,i}}\big(z_k - \textstyle\sum_{j=1}^{q-1} c_{k,j} z_{k-j}\big),
\end{aligned}
\tag{7}
$$

which turns out to be MPE, with the slight difference of taking the weighted sum of $\{z_j\}_{j=k-q+1}^k$ as opposed to the weighted sum of $\{z_j\}_{j=k-q}^{k-1}$ (See appendix for more details of MPE). Note that if the coefficients $c$ is computed in the following way: $b \in \mathrm{argmin}_{a \in \mathbb{R}^{q+1}, \sum_j a_j = 1} \|\sum_{j=0}^q a_j v_{k-j}\|$ and $b_0 \neq 0$ and define $c_j \overset{\text{def}}{=} -b_j/b_0$ for $j = 1, \ldots, q$. Then $(1 - \sum_{i=1}^q c_i)^{-1} = \frac{b_0}{b_0 + \sum_{j=1}^q b_j} = b_0$, and $\bar{z}_{k,\infty} = \sum_{j=0}^{q-1} b_j z_{k-j}$ is precisely the RRE update (again with the slight difference of summing over iterates shifted by one iteration).

**Acceleration guarantee for A³DMM** Let $\{z_k\}_{k\in\mathbb{N}}$ be a sequence in $\mathbb{R}^n$ and let $v_k \overset{\text{def}}{=} z_k - z_{k-1}$. Assume that $v_k = M v_{k-1}$ for some $M \in \mathbb{R}^{n\times n}$. Denote $\lambda(M)$ the spectrum of $M$. The following proposition provides control on the extrapolation error for $\bar{z}_{k,s}$ from (6).

**Proposition 4.3.** *Define the coefficient fitting error by* $\epsilon_k \overset{\text{def}}{=} \min_{c\in\mathbb{R}^q} \|V_{k-1}c - v_k\|$.

(i) *For* $s \in \mathbb{N}$, *we have*

$$\|\bar{z}_{k,s} - z^\star\| \leq \|z_{k+s} - z^\star\| + B_s \epsilon_k,
\tag{8}$$

*where* $B_s \overset{\text{def}}{=} \sum_{\ell=1}^s \|M^\ell\| |\sum_{i=0}^{s-\ell} (C_k^i)_{(1,1)}|$. *If* $\rho(M) < 1$ *and* $\rho(C_k) < 1$, *then* $\sum_i c_{k,i} \neq 1$ *and* $B_s$ *is uniformly bounded in* $s$. *For* $s = +\infty$, $B_\infty \overset{\text{def}}{=} |1 - \sum_i c_{k,i}|^{-1} \sum_{\ell=1}^\infty \|M\|^\ell$

(ii) *Suppose that* $M$ *is diagonalizable. Let* $(\lambda_j)_j$ *denote its distinct eigenvalues ordered such that* $|\lambda_j| \geq |\lambda_{j+1}|$ *and* $|\lambda_1| = \rho(M) < 1$. *Suppose that* $|\lambda_q| > |\lambda_{q+1}|$.

  - *Asymptotic bound (fixed* $q$ *and as* $k \to +\infty$): $\epsilon_k = \mathcal{O}(|\lambda_{q+1}|^k)$.
  - *Non-asymptotic bound (fixed* $q$ *and* $k$): *Suppose* $\lambda(M)$ *is real-valued and contained in* $[\alpha, \beta]$ *with* $-1 < \alpha < \beta < 1$. *Then, let* $K \overset{\text{def}}{=} 2\|z_0 - z^\star\| \|(\mathrm{Id}-M)^{\frac{1}{2}}\|$ *and* $\eta = \frac{1-\alpha}{1-\beta}$

$$\frac{\epsilon_k}{1 - \sum_i c_{k,i}} \leq K\beta^{k-q}\big(\tfrac{\sqrt{\eta}-1}{\sqrt{\eta}+1}\big)^q.
\tag{9}$$

**Remark 4.4.**

- From Theorem 2.2(ii), when $R$ and $J$ are both polyhedral, we have a perfect local linearisation with the corresponding linearisation matrix being normal and hence, the conditions of Proposition 4.3 holds for all $k$ large enough. The first bound (i) shows that the extrapolated point $\bar{z}_{k,s}$ moves along the true trajectory as $s$ increases, up to the fitting error $\epsilon_k$. Although $\bar{z}_{k,\infty}$ is essentially an MPE update which is known to satisfy error bound (9) (see [28]), this proposition offers a further interpretation of these extrapolation methods in terms of following the "sequence trajectory", and combined with our local analysis of ADMM, provides justification of these methods for the acceleration of non-smooth optimisation problems.
- Proposition 4.3 (ii) shows that extrapolation improves the convergence rate from $\mathcal{O}(|\lambda_1|^k)$ to $\mathcal{O}(|\lambda_{q+1}|^k)$, and the nonasymptotic bound shows that the improvement of extrapolation is optimal in the sense of Nesterov [22]. Recalling the form of the eigenvalues of $M$ from Theorem 2.2, in the case of two nonsmooth polyhedral terms, we must have $|\lambda_{2j-1}| = |\lambda_{2j}| > |\lambda_{2j+1}|$ for all $j \geq 1$. Hence, no acceleartion can be guaranteed or observed when $q = 1$, while the choice of $q = 2$ provides guaranteed acceleration.

Extension of A$^3$DMM to variants of ADMM is provided in Section B of the supplementary material.

# 5   Numerical experiments

Below we present numerical experiments on affine constrained minimisation (*e.g.* Basis Pursuit) and LASSO problems to demonstrate the performance of A$^3$DMM. Extra comparisons can be found in the supplementary material Section C. In the numerical comparison below, we mainly compare with the original ADMM and its inertial version (3) with fixed $a_k \equiv 0.3$. For the proposed A$^3$DMM, two settings are considered: $(q, s) = (4, 100)$ and $(q, s) = (4, +\infty)$. MATLAB source codes for reproducing the results can be found at: https://github.com/jliang993/A3DMM.

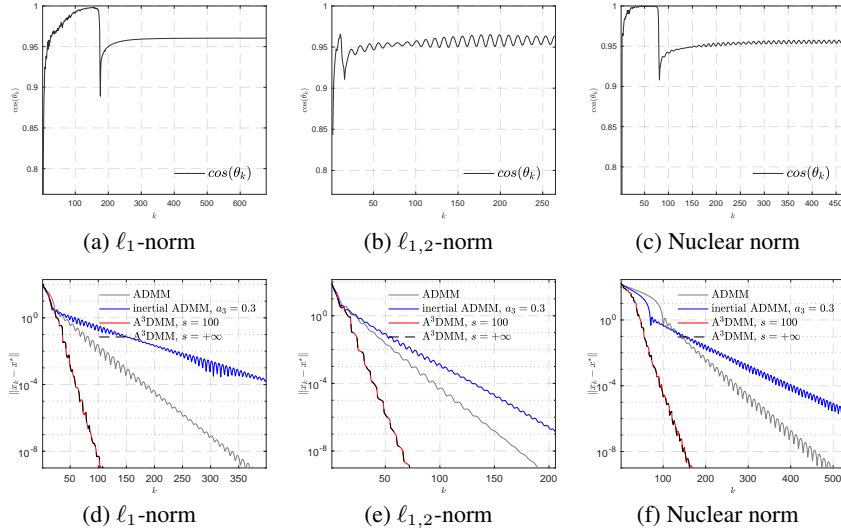

Figure 2: Performance comparisons and $\{\theta_k\}_{k \in \mathbb{N}}$ of ADMM for affine constrained problem.

**Affine constrained minimisation**   Consider the following constrained problem, given $\mathring{x}$

$$\min_{x \in \mathbb{R}^n} R(x) \quad \text{such that} \quad Kx = K\mathring{x}. \tag{10}$$

Denote the set $\Omega \stackrel{\text{def}}{=} \{x \in \mathbb{R}^n : Kx = K\mathring{x}\}$ and $\iota_\Omega$ its indicator function. Then (10) can be written as

$$\min_{x, y \in \mathbb{R}^n} R(x) + \iota_\Omega(y) \quad \text{such that} \quad x - y = 0, \tag{11}$$

which is special case of $(\mathcal{P}_{\text{ADMM}})$ with $A = \text{Id}$, $B = -\text{Id}$ and $b = 0$. Here $K$ is generated from the standard Gaussian ensemble, and the following three choices of $R$ are considered:

$\ell_1$**-norm** $(m, n) = (640, 2048)$, $\mathring{x}$ is 128-sparse;
$\ell_{1,2}$**-norm** $(m, n) = (640, 2048)$, $\mathring{x}$ has 32 non-zero blocks of size 4;
**Nuclear norm** $(m, n) = (1448, 64 \times 64)$, $\mathring{x}$ has rank of 4.

The property of $\{\theta_k\}_{k \in \mathbb{N}}$ is shown in Figure 2 (a)-(c). Note that the indicator function $\iota_\Omega(y)$ in (11) is polyhedral since $\Omega$ is an affine subspace,

- As $\ell_1$-norm is polyhedral, we have in Figure 2(a) that $\theta_k$ is converging to a constant which complies with Theorem 2.2(ii).
- Since $\ell_{1,2}$-norm and nuclear norm are no longer polyhedral functions, we have that $\theta_k$ eventually oscillates in a range, meaning that the trajectory of $\{z_k\}_{k\in\mathbb{N}}$ is an elliptical spiral.

Comparisons of the four schemes are shown below in Figure 2 (d)-(f):

- Since both functions in (11) are non-smooth, the eventual trajectory of $\{z_k\}_{k\in\mathbb{N}}$ for ADMM is spiral. Inertial ADMM fails to provide acceleration locally.
- A$^3$DMM is faster than both ADMM and inertial ADMM. For the two different settings of A$^3$DMM, their performances are very close.

**LASSO** We consider again the LASSO problem (4) with three datasets from LIBSVM[2]. The numerical experiments are provided below in Figure 3.

It can be observed that the proposed A$^3$DMM is significantly faster than the other schemes, especially for $s = +\infty$. Between ADMM and inertial ADMM, the inertial technique can provided consistent acceleration for all three examples since $\theta_k \to 0$; See first row of Figure 3. For Figure 3 (a), the oscillation after $k = 2000$ is due to machine error.

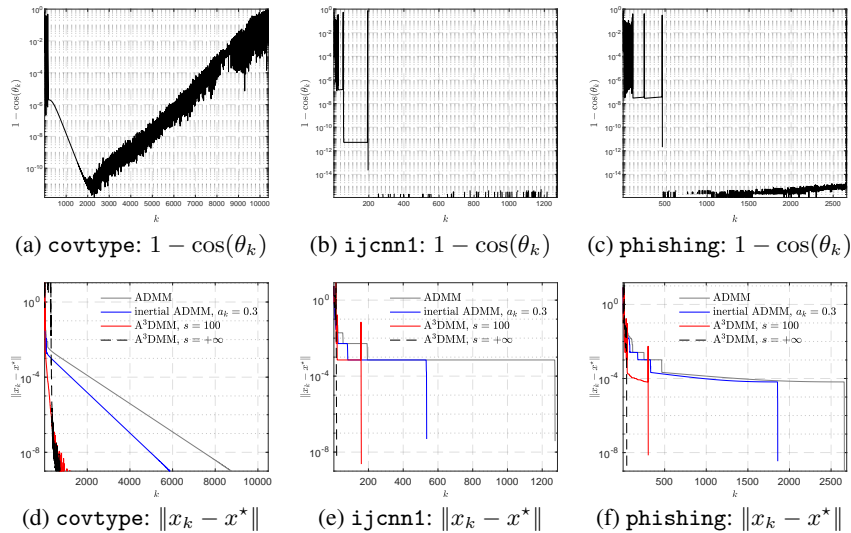

(a) `covtype`: $1 - \cos(\theta_k)$  (b) `ijcnn1`: $1 - \cos(\theta_k)$  (c) `phishing`: $1 - \cos(\theta_k)$

(d) `covtype`: $\|x_k - x^\star\|$  (e) `ijcnn1`: $\|x_k - x^\star\|$  (f) `phishing`: $\|x_k - x^\star\|$

Figure 3: Performance comparisons for LASSO problem.

## 6 Conclusions

In this article, by analyzing the trajectory of the fixed point sequences associated to ADMM and extrapolating along the trajectory, we provide an alternative derivation of these methods. Furthermore, our local linear analysis allows for the application of previous results on extrapolation methods, and hence provides guaranteed (local) acceleration.

### Acknowledgments

We would like to thank Arieh Iserles for pointing out the connection between trajectory following adaptive acceleration and vector extrapolation. We also like to thank the reviewers whose comments helped to improve the paper. JL was partly supported by Leverhulme trust and Newton trust.

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
