[Supplementary Material · supplementary.pdf]

# Supplementary Material for
## *Trajectory of Alternating Direction Method of Multipliers and Adaptive Acceleration*

**Clarice Poon**
University of Bath, Bath UK
cmhsp20@bath.ac.uk

**Jingwei Liang**
University of Cambridge, Cambridge UK
jl993@cam.ac.uk

## Abstract

The organization of the supplementary material is as follows: In Section A, more substantial discussions on the failure of inertia are provided. Variants of ADMM, including relaxed ADMM and symmetric ADMM, are discussed in Section B. In Section C, we provide more numerical experiments to demonstrate the performance of A$^3$DMM. The proofs of the main results of the paper are contained in Sections D, E and F, where in Section D some preliminary results on angles between subspaces and Riemannian geometry are provide, in Section E the proofs for the trajectory of ADMM are provided, and lastly in in Section F we provide proofs for A$^3$DMM.

## A    The failure of inertial acceleration continue

In this part, to support the discussion of Section 3, we provide extra discussion on why inertial acceleration, in particular Nesterov/FISTA, will fail when the (leading) eigenvalue of $M$ is complex.

Let $M \in \mathbb{R}^{n \times n}$ be a square matrix and consider the following linear equation

$$z_{k+1} = M z_k. \tag{A.1}$$

According to [31], (A.1) is linearly convergent when the spectral radius of $M$ is strictly smaller than 1, *i.e.* $\rho(M) < 1$. For simplicity, consider the inertial version of (A.1) with fixed inertial parameter $a_k \equiv a \in [0, 1]$, we get

$$
\begin{aligned}
y_k &= z_k + a(z_k - z_{k-1}) \\
z_{k+1} &= M y_k.
\end{aligned}
\tag{A.2}
$$

The above scheme corresponds to the local linearization of the inertial ADMM (3) without the small $o$-term. Define the augmented variable $w_k = \begin{pmatrix} z_k \\ z_{k-1} \end{pmatrix}$ and block matrix $\widetilde{M} \overset{\text{def}}{=} \begin{bmatrix} (1+a)M & -aM \\ \text{Id} & 0 \end{bmatrix}$, then (A.2) can be written as

$$w_{k+1} = \widetilde{M} w_k. \tag{A.3}$$

To guarantee the convergence of (A.3), we require the spectral radius satisfying $\rho(\widetilde{M}) < 1$. Therefore, in the following, motivated by [31, 24, 26], we discuss the property of the spectral radius $\rho(\widetilde{M})$ and the conditions such that $\rho(\widetilde{M}) < 1$.

Let $\eta, \rho$ be the leading eigenvalues of $M$ and $\widetilde{M}$, respectively. According to [26, Proposition 4.6], we have the following lemma regarding the relation between $\eta$ and $\rho$.

**Lemma A.1 ([26, Proposition 4.6]).** *Suppose $\begin{pmatrix} r_1 \\ r_2 \end{pmatrix}$ is the eigenvector of $\widetilde{M}$ corresponding to eigenvalue $\rho$, then it must satisfy $r_1 = \rho r_2$. Moreover, $r_2$ is an eigenvector of $M$ associated to*

*eigenvalue $\eta$, where $\eta$ and $\rho$ satisfy the relation*

$$\rho^2 - (1+a)\eta\rho + a\eta = 0. \tag{A.4}$$

The relation (A.4) is a simple quadratic equation of $\rho$, we have

$$\rho = \frac{(1+a)\eta + \sqrt{(1+a)^2\eta^2 - 4a\eta}}{2}. \tag{A.5}$$

The value of $|\rho|$ depends on $a$ and $\eta$, and the discussion splits into two scenarios: $\eta$ is real and $\eta$ is complex.

## A.1 Real $\eta$

When $\eta$ is real valued, the property of $\rho$ is well studied, we refer to [26] and references therein for detailed discussions. Basically, we have that

$$|\rho| = \begin{cases} (1+a)^2\eta^2 \geq 4a\eta : \rho \text{ is real}, |\rho| < 1 \text{ holds for any } a \in [0,1], \\ (1+a)^2\eta^2 < 4a\eta : \rho \text{ is complex}, |\rho| = \sqrt{a\eta} < 1 \text{ holds for any } a \in [0,1]. \end{cases}$$

The above result can be summarized below.

**Lemma A.2 ([26, Proposition 4.6]).** *Given any $a \in [0,1]$, we have $|\rho| < 1$ as long as $0 \leq \eta < 1$.*

To demonstrate the above result, we consider fixing $\eta$ and varying $a \in [0,1]$. Two choices of $\eta$ are considered $\eta = 0.9, 0.98$, the value of $|\rho|$ is plotted in Figure A.1 in black line. It can be observed that $|\rho|$ is strictly smaller than one for both choices of $\eta$. Note that $|\rho|$ reaches a minimal value for some $a$, we refer to [26] for detailed discussion on this.

Figure A.1: The value of $|\rho|$ under fixed $|\eta|$ and $a \in [0,1]$.

## A.2 Complex $\eta$

When $\eta$ is complex, it can be written as $\eta = |\eta|e^{i\alpha}$ where $\alpha$ is the argument of $\eta$. The dependence of $|\rho|$ on $a$ and $\eta$ becomes much more complicated, below we briefly demonstrate numerically the properties of $|\rho|$.

**General form** $\eta = |\eta|e^{i\alpha}$  For this case, we have

$$\rho = \frac{(1+a)\eta + \sqrt{(1+a)^2\eta^2 - 4a\eta}}{2} = \frac{(1+a)|\eta|e^{i\alpha} + \sqrt{(1+a)^2|\eta|^2 e^{i2\alpha} - 4a|\eta|e^{i\alpha}}}{2}.$$

Suppose $(x + iy)^2 = (1+a)^2|\eta|^2 e^{i2\alpha} - 4a|\eta|e^{i\alpha}$, we get

$$x^2 - y^2 = (1+a)^2|\eta|^2\cos(2\alpha) - 4a|\eta|\cos(\alpha)$$

$$xy = \frac{(1+a)^2|\eta|^2\sin(2\alpha) - 4a|\eta|\sin(\alpha)}{2},$$

which can be simplified to a equation of $x$

$$x^4 - \left((1+a)^2|\eta|^2\cos(2\alpha) - 4a|\eta|\cos(\alpha)\right)x^2 - \frac{((1+a)^2|\eta|^2\sin(2\alpha) - 4a|\eta|\sin(\alpha))^2}{4} = 0.$$

Solving the above equation, we get

$$x = \left(\frac{((1+a)^2|\eta|^2\cos(2\alpha)-4a|\eta|\cos(\alpha))+\sqrt{((1+a)^2|\eta|^2\cos(2\alpha)-4a|\eta|\cos(\alpha))^2+((1+a)^2|\eta|^2\sin(2\alpha)-4a|\eta|\sin(\alpha))^2}}{2}\right)^{1/2},$$

$$y = \frac{(1+a)^2|\eta|^2\sin(2\alpha) - 4a|\eta|\sin(\alpha)}{2x},$$

here we only take the positive root $x$. Back to the expression of $\rho$, we get

$$\rho = \frac{(1+a)|\eta|e^{i\alpha} + (x+iy)}{2} = \frac{((1+a)|\eta|\cos(\alpha)+x) + i((1+a)|\eta|\sin(\alpha)+y)}{2}.$$

Given the complicated form of $x$, the analysis of $|\rho|$ becomes rather difficult. Therefore, below we discuss the properties of $|\rho|$ through numerical verification.

Similar to the real $\eta$ case, $|\eta| = 0.9, 0.98$ are considered. Let $\alpha$ be the argument of $\eta$, then we have $\eta = |\eta|e^{i\alpha}$. In total, six choices of $\alpha$ are considered: $\alpha \in \{\frac{\pi}{4}, \frac{\pi}{8}, \frac{\pi}{16}, \frac{\pi}{32}, \frac{\pi}{64}, \frac{\pi}{128}\}$. The value of $|\rho|$ are shown in Figure A.1. Taking Figure A.1 (a) for example, we have the following observations: let $a_\alpha$ be the largest $a$ allowed such that $|\rho| \le 1$,

- For all choices of $\alpha$ except $\alpha = \frac{\pi}{128}$, we have $a_\alpha < 1$.
- The larger the value of $\alpha$, the smaller the value of $a_\alpha$, see the green line in both figures.

From the above discussion, we can conclude that

- The inertial scheme is robust when all the eigenvalues of $M$ are real, and we can afford the inertial parameter up to 1 which includes the FISTA [5] schemes as $a_k \to 1$, same for the Nesterov's accelerated gradient descent.
- When $M$ has complex eigenvalue(s), which is not necessary to the leading eigenvalue, the largest value of $a$ such that $|\rho| < 1$ is smaller than 1 and FISTA/Nesterov's scheme will fail.

To complete the discussion, we consider the values of $|\rho|$ under $\alpha \in [0, \pi/2]$ and $a \in [0, 1]$. The results are shown below in Figure A.2. Again $|\eta| = 0.9, 0.98$ are considered. The horizontal axis is for $\alpha$ while the vertical is for $a$, each point inside the square stands for the value of $|\rho|$ with colorbar provided. In each figure:

- The *red* line stands for $|\rho| = 1$. Therefore, only for the area below the red line we have $|\rho| < 1$. Given any $\alpha \in [0, \pi/2]$, the larger the value of $\alpha$, the smaller range of choice of $a$ such that $|\rho| < 1$. This coincides with the observations from Figure A.1.
- The *magenta* line stands for $|\rho| = |\eta|$. Only the small area below the magenta line has $|\rho| < |\eta|$, meaning that acceleration can be obtained. As a result, given $\eta = |\eta|e^{i\alpha}$, when $\alpha$ is large enough, such as about $\pi/8$ for $|\eta| = 0.9$, inertial will fail to provide acceleration.

(a) $|\eta| = 0.9$  (b) $|\eta| = 0.98$

Figure A.2: The value of $|\rho|$ under fixed $\eta$ and $a \in [0, 1]$.

It should be noted that, for the above discussion, we consider the case that the leading eigenvalue is *complex*, while the rest of the eigenvalues are *real*. For the case leading eigenvalue is *real* while

the rest are *complex*, then the spectral radius of $\widetilde{M}$ will be determined by the non-leading complex eigenvalues when the inertial parameter $a$ is large enough. Consequently, the FISTA inertial parameter rule still can not be applied, unless the magnitude of the leading eigenvalue is small enough; See Figure A.2 (a).

(a) $\alpha \in \left\{ \frac{\pi}{4}, \frac{\pi}{8}, \frac{\pi}{16}, \frac{\pi}{32}, \frac{\pi}{64}, \frac{\pi}{128} \right\}$

(b) $\alpha \in [0, \frac{\pi}{2}]$

Figure A.3: The value of $|\rho|$ when $\eta = \cos(\alpha) \mathrm{e}^{\mathrm{i}\alpha}$ and $a \in [0, 1]$.

**Special case** $\eta = \cos \alpha e^{\mathrm{i}\alpha}$   Now we consider a special case where $\eta = \cos(\alpha) e^{\mathrm{i}\alpha}, \alpha \in [0, \pi/2]$ which corresponds to the case $R, J$ in $(\mathcal{P}_{\mathrm{ADMM}})$ are locally polyhedral around $x^\star, y^\star$. Similar to above, six choices of $\alpha$ are considered: $\alpha \in \left\{ \frac{\pi}{4}, \frac{\pi}{8}, \frac{\pi}{16}, \frac{\pi}{32}, \frac{\pi}{64}, \frac{\pi}{128} \right\}$. The value of $|\rho|$ is shown below in Figure A.3 (a). It can be observed that, for each $\alpha$, the value of $|\rho|$ is monotonically increasing as the value of $a$ increases, which means *inertial slows down the speed of convergence*. In Figure A.3 (b), we consider the value of $|\rho|$ under $\alpha \in [0, \pi/2]$ and $a \in [0, 1]$. We have

- Similar to Figure A.2, the *red line* stands for $|\rho| = 1$. For each $\alpha$, $|\rho| < 1$ for all the choices of $a$ under the red line.
- The *magenta* line stands for $|\rho| = |\eta|$. It can be observed that, except for $\alpha = 0$ where $|\rho| = 1$ holds for all $a \in [0, 1]$, $|\rho| = 1$ holds only for $a = 0$ when $\alpha \in ]0, \pi/2]$.

Therefore, we can conclude that when $R, J$ are locally polyhedral around the solution $x^\star, y^\star$, inertial scheme will not provide any acceleration.

# B   Discussions

In this section, we discuss two variants of ADMM: relaxed ADMM and symmetric ADMM are discussed, and then build connections between ADMM and Douglas–Rachford [14] and Peaceman–Rachford splitting [30] methods.

## B.1   Variants of ADMM

**Relaxed ADMM**   In the literature, a popular variant of ADMM is the *relaxed ADMM* which takes the following iteration procedure:

$$
\begin{aligned}
x_k &= \operatorname{argmin}_{x \in \mathbb{R}^n} R(x) + \tfrac{\gamma}{2} \| Ax + By_{k-1} - b + \tfrac{1}{\gamma} \psi_{k-1} \|^2, \\
\bar{x}_k &= \phi A x_k - (1 - \phi)(By_{k-1} - b), \\
y_k &= \operatorname{argmin}_{y \in \mathbb{R}^m} J(y) + \tfrac{\gamma}{2} \| \bar{x}_k + By - b + \tfrac{1}{\gamma} \psi_{k-1} \|^2, \\
\psi_k &= \psi_{k-1} + \gamma(\bar{x}_k + By_k - b),
\end{aligned}
\tag{B.1}
$$

where $\phi \in [0, 2]$ is the relaxation parameter.

In its dual form, the relaxed ADMM is equivalent to the *relaxed* Douglas–Rachford splitting applied to solve $(\mathcal{D}_{\mathrm{ADMM}})$, see Section B.2.1. The convergence of (B.1) can be guaranteed for $\phi \in ]0, 2[$ [2]. Similar to (2), define $z_k \overset{\text{def}}{=} \psi_{k-1} + \gamma \bar{x}_k = \psi_{k-1} + \gamma(\phi A x_k - (1 - \phi)(By_{k-1} - b))$, we can rewrite

the relaxed ADMM into the following form

$$
\begin{aligned}
x_k &= \operatorname{argmin}_{x\in\mathbb{R}^n} R(x) + \tfrac{\gamma}{2}\|Ax - \tfrac{1}{\gamma}(z_{k-1} - 2\psi_{k-1})\|^2, \\
z_k &= \psi_{k-1} + \gamma(\phi Ax_k - (1-\phi)(By_{k-1} - b)), \\
y_k &= \operatorname{argmin}_{y\in\mathbb{R}^m} J(y) + \tfrac{\gamma}{2}\|By + \tfrac{1}{\gamma}(z_k - \gamma b)\|^2, \\
\psi_k &= z_k + \gamma(By_k - b).
\end{aligned}
\tag{B.2}
$$

**Symmetric ADMM**  As aforementioned, see also Section B.2.1, the ADMM iteration (1) is equivalent to applying Douglas–Rachford splitting to the dual problem $(\mathcal{D}_{\mathrm{ADMM}})$ [17]. It is also pointed out in [17] that, if the Peaceman–Rachford splitting method [30] is applied to solve $(\mathcal{D}_{\mathrm{ADMM}})$, then it leads to the following iteration in the primal form

$$
\begin{aligned}
x_k &= \operatorname{argmin}_{x\in\mathbb{R}^n} R(x) + \tfrac{\gamma}{2}\|Ax + By_{k-1} - b + \tfrac{1}{\gamma}\psi_{k-1}\|^2, \\
\psi_{k-\frac{1}{2}} &= \psi_{k-1} + \gamma(Ax_k + By_{k-1} - b), \\
y_k &= \operatorname{argmin}_{y\in\mathbb{R}^m} J(y) + \tfrac{\gamma}{2}\|Ax_k + By - b + \tfrac{1}{\gamma}\psi_{k-\frac{1}{2}}\|^2, \\
\psi_k &= \psi_{k-\frac{1}{2}} + \gamma(Ax_k + By_k - b),
\end{aligned}
\tag{B.3}
$$

which is also called the *symmetric ADMM*. A brief derivation is provided below in Section B.2.2, and we refer to [17, 22] and the references therein for more detailed discussions.

In general, the conditions needed for the convergence of (B.3) is stronger than the standard ADMM (1), which is due to the fact that stronger conditions are needed to guarantee the convergence of Peaceman–Rachford splitting method [17]. However, when (B.3) converges, it tends to provide faster performance than (1) [17]. Similar to (2), if we define $z_k = \psi_k - \gamma By_k + \gamma b = \psi_{k-\frac{1}{2}} + \gamma Ax_k$, then (B.3) is equivalent to

$$
\begin{aligned}
x_k &= \operatorname{argmin}_{x\in\mathbb{R}^n} R(x) + \tfrac{\gamma}{2}\|Ax + \tfrac{1}{\gamma}(2\psi_{k-1} - z_{k-1})\|^2, \\
z_k &= \psi_{k-1} + \gamma(2Ax_k + By_{k-1} - b), \\
y_k &= \operatorname{argmin}_{y\in\mathbb{R}^m} J(y) + \tfrac{\gamma}{2}\|By + \tfrac{1}{\gamma}(z_k - \gamma b)\|^2, \\
\psi_k &= z_k + \gamma(By_k - b),
\end{aligned}
\tag{B.4}
$$

which can be written as the fixed-point iteration in terms of $z_k$, see Section B.2.2.

**Extension of A$^3$DMM to the variants**  We can summarize the standard (2), relaxed (B.1) and symmetric (B.4) ADMM into the following form

$$
\begin{aligned}
x_k &= \operatorname{argmin}_{x\in\mathbb{R}^n} R(x) + \tfrac{\gamma}{2}\|Ax + \tfrac{1}{\gamma}(2\psi_{k-1} - z_{k-1})\|^2, \\
z_k &= \mathcal{Z}(\gamma, \phi; x_k, y_{k-1}, \psi_{k-1}), \\
y_k &= \operatorname{argmin}_{y\in\mathbb{R}^m} J(y) + \tfrac{\gamma}{2}\|By + \tfrac{1}{\gamma}(z_k - \gamma b)\|^2, \\
\psi_k &= z_k + \gamma(By_k - b),
\end{aligned}
\tag{B.5}
$$

where $\mathcal{Z}$ represent the way of updating $z_k$; See (2), (B.1) and (B.4). Accordingly, we can easily adapt Algorithm 1 to the relaxed and symmetric ADMM, that is changing the update of $z_k$.

In Algorithm 1, we change the order of updates so that the extrapolation step only needs to be carried out on $z_k$. This is due to the fact, the update of $y_k$ only depends on $z_k$, and such an arrangement requires the minimal computational overhead.

## B.2  Fixed-point characterization and convergence of ADMM

We discuss the relation between ADMM and Douglas–Rachford splitting [14] and Peaceman–Rachford splitting [30].

### B.2.1  Relaxed ADMM and Douglas–Rachford splitting

It is well-known that ADMM is equivalent to applying Douglas–Rachford splitting [14] to solve the dual problem of $(\mathcal{P}_{\mathrm{ADMM}})$ which reads

$$
\max_{\psi\in\mathbb{R}^p} -\big(R^*(-A^T\psi) + J^*(-B^T\psi) + \langle\psi,\, b\rangle\big),
\tag{$\mathcal{D}_{\mathrm{ADMM}}$}
$$

where $R^*(v) \stackrel{\text{def}}{=} \sup_{x \in \mathbb{R}^n} (\langle x, v \rangle - R(x))$ is called the Fenchel conjugate, or simply conjugate, of $R$. Below we first recall the equivalence between ADMM and Douglas–Rachford which was first established in [17], and then use the convergence of Douglas–Rachford splitting which is well established in the literature [2] to conclude the convergence of ADMM.

Consider the relaxed ADMM (B.1) , when $\phi = 1$, the relaxed ADMM recovers the standard ADMM (2). Below show demonstrate that the relaxed ADMM is equivalent to the relaxed Douglas–Rachford applying to solve $(\mathcal{D}_{\text{ADMM}})$.

- Define $z_k = \psi_k - \gamma(By_k - b)$, we have

$$
\begin{aligned}
z_k = \psi_k - \gamma By_k + \gamma b &= \psi_{k-1} + \gamma \bar{x}_k \\
&= \phi \psi_{k-1} + \phi \gamma A x_k + (1 - \phi)\psi_{k-1} - (1 - \phi)\gamma(By_{k-1} - b) \\
&= (1 - \phi)z_{k-1} + \phi(\psi_{k-1} + \gamma A x_k) \\
&= (1 - \phi)z_{k-1} + \phi(z_{k-1} + u_k - \psi_{k-1}).
\end{aligned}
$$

  When $\phi = 1$, we have $z_k = \psi_{k-1} + \gamma A x_k$.

- For the update of $x_k$, denote $u_k = \psi_{k-1} + \gamma(Ax_k + By_{k-1} - b)$. Since $A$ has full column rank, we have $x_k$ is the unique minimiser of $R(x) + \frac{\gamma}{2}\|Ax + By_{k-1} - b + \frac{1}{\gamma}\psi_{k-1}\|^2$. Let $R^*$ be the conjugate of $R$, then owing to duality, we get

$$
\begin{aligned}
&x_k = \operatorname{argmin}_{x \in \mathbb{R}^n} R(x) + \tfrac{\gamma}{2}\|Ax + By_{k-1} - b + \tfrac{1}{\gamma}\psi_{k-1}\|^2 \\
\Longleftrightarrow\ &0 \in \partial R(x_k) + \gamma A^T(Ax_k + By_{k-1} - b + \tfrac{1}{\gamma}\psi_{k-1}) \\
\Longleftrightarrow\ &- A^T u_k \in \partial R(x_k) \\
\Longleftrightarrow\ &x_k \in \partial R^*(-A^T u_k) \\
\Longleftrightarrow\ &u_k - \gamma A x_k \in u_k + \gamma \partial (R^* \circ -A^T)(u_k) \\
\Longleftrightarrow\ &u_k = (\operatorname{Id} + \gamma \partial(R^* \circ -A^T))^{-1}(u_k - \gamma A x_k) \\
\Longleftrightarrow\ &u_k = (\operatorname{Id} + \gamma \partial(R^* \circ -A^T))^{-1}(2\psi_{k-1} - z_{k-1}).
\end{aligned}
$$

- For the update of $y_k$, the full column rank of $B$ also ensures that $y_k$ is the unique minimiser of $J(y) + \frac{\gamma}{2}\|\bar{x}_k + By - b + \frac{1}{\gamma}\psi_{k-1}\|^2$. Since $\psi_k = \psi_{k-1} + \gamma(\bar{x}_k + By_k - b)$, then

$$
\begin{aligned}
&y_k = \operatorname{argmin}_{y \in \mathbb{R}^m} J(y) + \tfrac{\gamma}{2}\|\bar{x}_k + By - b + \tfrac{1}{\gamma}\psi_{k-1}\|^2 \\
\Longleftrightarrow\ &0 \in \partial J(y_k) + \gamma B^T(\bar{x}_k + By_k - b + \tfrac{1}{\gamma}\psi_{k-1}) \\
\Longleftrightarrow\ &- B^T \psi_k \in \partial J(y_k) \\
\Longleftrightarrow\ &y_k \in \partial J^*(-B^T \psi_k) \\
\Longleftrightarrow\ &\psi_k - \gamma By_k \in \psi_k + \gamma \partial(J^* \circ -B^T)(\psi_k) \\
\Longleftrightarrow\ &\psi_k = (\operatorname{Id} + \gamma \partial(J^* \circ -B^T))^{-1}(\psi_k - \gamma By_k) \\
\Longleftrightarrow\ &\psi_k = (\operatorname{Id} + \gamma \partial(J^* \circ -B^T))^{-1}(z_k - \gamma b).
\end{aligned}
$$

- Combining all the relations we get

$$
\begin{aligned}
u_k &= (\operatorname{Id} + \gamma \partial(R^* \circ -A^T))^{-1}(2\psi_{k-1} - z_{k-1}), \\
z_k &= (1 - \phi)z_{k-1} + \phi(z_{k-1} + u_k - \psi_{k-1}), \\
\psi_k &= (\operatorname{Id} + \gamma \partial(J^* \circ -B^T))^{-1}(z_k - \gamma b),
\end{aligned} \tag{B.6}
$$

  which is exactly the iteration of Douglas–Rachford splitting applied to solve the dual $(\mathcal{D}_{\text{ADMM}})$.

Define the operators

$$
\mathcal{F}_{\text{DR}} \stackrel{\text{def}}{=} \tfrac{1}{2}\operatorname{Id} + \tfrac{1}{2}\left(2(\operatorname{Id} + \gamma \partial(R^* \circ -A^T))^{-1} - \operatorname{Id}\right)\left(2(\operatorname{Id} + \gamma \partial(J^* \circ -B^T))^{-1} - \operatorname{Id}\right)
$$

and $\mathcal{F}_{\text{DR}}^\phi = (1 - \phi)\operatorname{Id} + \phi \mathcal{F}_{\text{DR}}$, then (B.6) can be written as the fixed-point iteration in terms of $z_k$

$$
z_k = \mathcal{F}_{\text{DR}}^\phi(z_{k-1}).
$$

It should be noted that for $z_k$ we have $z_k = \psi_k - \gamma B y_k + \gamma b = \psi_{k-1} + \gamma A x_k$ which is the same as in (2). Owing to [2], that $\mathcal{F}_{\mathrm{DR}}^\phi$ is averaged non-expansive with the set of fixed-points $\mathrm{fix}(\mathcal{F}_{\mathrm{DR}})$ being non-empty, and there exists a fixed-point $z^\star \in \mathrm{fix}(\mathcal{F}_{\mathrm{DR}})$ such that $z_k \to z^\star$ which concludes the convergence of $\{z_k\}_{k \in \mathbb{N}}$. Then we have $u_k, \psi_k$ converging to $\psi^\star = (\mathrm{Id} + \gamma \partial(J^* \circ -B^T))^{-1}(z^\star - \gamma b)$ which is a dual solution of the problem $(\mathcal{D}_{\mathrm{ADMM}})$. The convergence of the primal ADMM sequences $\{x_k\}_{k \in \mathbb{N}}$ and $\{y_k\}_{k \in \mathbb{N}}$ follows immediately.

Owing to the above equivalence between ADMM and Douglas–Rachford splitting, we get the following relations

$$
\begin{aligned}
\|z_k - z_{k-1}\| &\leq \|z_{k-1} - z_{k-2}\|, \\
\|\psi_k - \psi_{k-1}\| &\leq \|z_k - z_{k-1}\| \leq \|z_{k-1} - z_{k-2}\|, \\
\|u_k - u_{k-1}\| &\leq \|2\psi_{k-1} - z_{k-1} - 2\psi_{k-2} + z_{k-2}\| \leq 3\|z_{k-1} - z_{k-2}\|, \\
\gamma\|Ax_k - Ax_{k-1}\| &\leq \|z_k - z_{k-1}\| + \|\psi_{k-1} - \psi_{k-2}\| \leq 2\|z_{k-1} - z_{k-2}\|, \\
\gamma\|By_k - By_{k-1}\| &\leq \|z_k - z_{k-1}\| + \|\psi_k - \psi_{k-1}\| \leq 2\|z_{k-1} - z_{k-2}\|,
\end{aligned}
\tag{B.7}
$$

which are needed in the proofs below.

### B.2.2 Symmetric ADMM and Peaceman–Rachford splitting

Below we present a short discussion on the relation between the symmetric ADMM and Peaceman–Rachford splitting method [30], which was first established in [17].

- For the update of $x_k$, let $u_k = \psi_{k-\frac{1}{2}} = \psi_{k-1} + \gamma(Ax_k + By_{k-1} - b)$ and $z_k = \psi_k - \gamma By_k + \gamma b$. As $A$ has full column rank, $x_k$ is the unique minimiser of $R(x) + \frac{\gamma}{2}\|Ax + By_{k-1} - b + \frac{1}{\gamma}\psi_{k-1}\|^2$. Then owing to duality,

$$
\begin{aligned}
x_k &= \mathrm{argmin}_{x \in \mathbb{R}^n}\, R(x) + \tfrac{\gamma}{2}\|Ax + By_{k-1} - b + \tfrac{1}{\gamma}\psi_{k-1}\|^2 \\
\iff\quad &- A^T u_k \in \partial R(x_k) \\
\iff\quad &x_k \in \partial R^*(-A^T u_k) \\
\iff\quad &u_k = \big(\mathrm{Id} + \gamma\partial(R^* \circ -A^T)\big)^{-1}(u_k - \gamma A x_k) \\
\iff\quad &u_k = \big(\mathrm{Id} + \gamma\partial(R^* \circ -A^T)\big)^{-1}(2\psi_{k-1} - z_{k-1}).
\end{aligned}
$$

- For $y_k$, the full column rank of $B$ ensures the uniqueness of $y_k$. Since $\psi_k = \psi_{k-\frac{1}{2}} + \gamma(Ax_k + By_k - b)$, then

$$
\begin{aligned}
y_k &= \mathrm{argmin}_{y \in \mathbb{R}^m}\, J(y) + \tfrac{\gamma}{2}\|Ax_k + By - b + \tfrac{1}{\gamma}\psi_{k-\frac{1}{2}}\|^2 \\
\iff\quad &- B^T \psi_k \in \partial J(y_k) \\
\iff\quad &y_k \in \partial J^*(-B^T \psi_k) \\
\iff\quad &\psi_k = \big(\mathrm{Id} + \gamma\partial(J^* \circ -B^T)\big)^{-1}(\psi_k - \gamma B y_k) \\
\iff\quad &\psi_k = \big(\mathrm{Id} + \gamma\partial(J^* \circ -B^T)\big)^{-1}(z_k - \gamma b).
\end{aligned}
$$

- For $z_k$, since $u_k = \psi_{k-\frac{1}{2}}$,

$$
z_k = \psi_k - \gamma B y_k + \gamma b = u_k + \gamma A x_k = 2u_k - \psi_{k-1} - \gamma(By_{k-1} - b) = z_{k-1} + 2(u_k - \psi_{k-1}).
$$

Combining the above relations we get

$$
\begin{aligned}
u_k &= \big(\mathrm{Id} + \gamma\partial(R^* \circ -A^T)\big)^{-1}(2\psi_{k-1} - z_{k-1}), \\
z_k &= z_{k-1} + 2(u_k - \psi_{k-1}), \\
\psi_k &= \big(\mathrm{Id} + \gamma\partial(J^* \circ -B^T)\big)^{-1}(z_k - \gamma b),
\end{aligned}
\tag{B.8}
$$

which is the iteration of Peaceman–Rachford splitting when applied to solve $(\mathcal{D}_{\mathrm{ADMM}})$.

Define the following operator

$$
\mathcal{F}_{\mathrm{PR}} = \Big(2\big(\mathrm{Id} + \gamma\partial(R^* \circ -A^T)\big)^{-1} - \mathrm{Id}\Big)\Big(2\big(\mathrm{Id} + \gamma\partial(J^* \circ -B^T)\big)^{-1} - \mathrm{Id}\Big),
$$

then (B.8) can be written as the fixed-point iteration in terms of $z_k$, that is

$$z_k = \mathcal{F}_{\mathrm{PR}}(z_{k-1}).$$

It should be noted that for $z_k$ we have $z_k = \psi_k - \gamma B y_k + \gamma b = \psi_{k-1} + \gamma A x_k$ which is the same as in (B.4). Different to the case of Douglas–Rachford, the operator $\mathcal{F}_{\mathrm{PR}}$ is only non-expansive [2], hence the conditions for $z_k$ to be convergent is stronger than that of $\mathcal{F}_{\mathrm{DR}}$. However, when it converges, it tends to be faster than Douglas–Rachford splitting [17].

## C  More numerical experiments

We present extra numerical experiments to demonstrate the performance of the proposed scheme. Same as Section 5, ADMM, inertial ADMM and two settings of A$^3$DMM are considered.

### C.1  Quadratic programming

Consider the following quadratic optimisation problem

$$\min_{x \in \mathbb{R}^n} \quad \tfrac{1}{2} x^T Q x + \langle q,\, x \rangle, \tag{C.1}$$
$$\text{such that} \quad x_i \in [\ell_i, r_i], \quad i = 1, ..., n.$$

Define the constraint set $\Omega = \{x \in \mathbb{R}^n : x_i \in [\ell_i, r_i], \quad i = 1, ..., n\}$, then (C.1) can be written as

$$\min_{x, y \in \mathbb{R}^n} \quad \tfrac{1}{2} x^T Q x + \langle q,\, x \rangle + \iota_\Omega(y) \quad \text{such that} \quad x - y = 0,$$

which is special case of $(\mathcal{P}_{\mathrm{ADMM}})$ with $A = \mathrm{Id}, B = -\mathrm{Id}$ and $b = 0$.

The angle $\theta_k$ of ADMM and the performances of the four schemes are provided in Figure (C.1), from which we observed that

- The angle $\theta_k$ is decreasing to $0$ at the beginning and then starts to increasing for $k \geq 2 \times 10^4$. This is mainly due to the fact that for $k \geq 2 \times 10^4$, the effects of machine error is becoming increasingly larger.
- Consistent with the observations in Section 5, the proposed A$^3$DMM schemes provides the best performance.

(a) Angle $\theta_k$        (b) Comparison of $\|x_k - x^\star\|$

Figure C.1: Performance comparisons and $\{\theta_k\}_{k \in \mathbb{N}}$ of ADMM for quadratic programming.

### C.2  Total variation based image inpainting

Now we consider a total variation (TV) based image inpainting problem. Let $u \in \mathbb{R}^{n \times n}$ be an image and $\mathcal{S} \in \mathbb{R}^{n \times n}$ be a Bernoulli matrix, the observation of $u$ under $\mathcal{S}$ is $f = \mathcal{P}_\mathcal{S}(u)$. The TV based image inpainting can be formulated as

$$\min_{x \in \mathbb{R}^{n \times n}} \|\nabla x\|_1 \quad \text{such that} \quad \mathcal{P}_\mathcal{S}(x) = f. \tag{C.2}$$

Define $\Omega \overset{\text{def}}{=} \{x \in \mathbb{R}^{n \times n} : \mathcal{P}_S(x) = f\}$, then (C.2) becomes

$$\min_{x \in \mathbb{R}^{n \times n}, y \in \mathbb{R}^{2n \times n}} \|y\|_1 + \iota_\Omega(x) \quad \text{such that} \quad \nabla x - y = 0, \tag{C.3}$$

which is special case of ($\mathcal{P}_{\text{ADMM}}$) with $A = \nabla, B = -\text{Id}$ and $b = 0$. For the update of $x_k$, we have from (2) that

$$x_k = \text{argmin}_{x \in \mathbb{R}^{n \times n}} \iota_\Omega(x) + \frac{\gamma}{2} \|\nabla x - \frac{1}{\gamma}(\bar{z}_{k-1} - 2\psi_{k-1})\|^2,$$

which does not admit closed form solution. In the implementation, finite-step FISTA is applied to roughly solve the above problem.

In the experiment, the `cameraman` image is used, and $50\%$ of the pixels is removed randomly. The angle $\theta_k$ of ADMM and the comparisons of the four schemes are provided in Figure C.2:

- Though both functions in (C.3) are polyhedral, since the subproblem of $x_k$ is solved approximately, the eventual angle actually is oscillating instead of being a constant.
- Inertial ADMM again is slower than the original ADMM as the trajectory of ADMM is a spiral.
- For the two A$^3$DMM schemes, their performances are close as previous examples.
- For PSNR the image quality assessment, Figure C.2(c) implies that A$^3$DMM is also the best.

(a) Angle $\{\theta_k\}_{k \in \mathbb{N}}$ of ADMM  (b) Comparison of $\|x_k - x^\star\|$  (c) PSNR value

Figure C.2: Property of $\{\theta_k\}_{k \in \mathbb{N}}$, performance comparison and image quality of ADMM for TV based image inpainting.

We also compare the visual quality of the images obtained by the four schemes for the 30'th iteration, which is shown below in Figure C.3. It can be observed that the image quality (2nd row of Figure C.3) is much better than the 1st row of ADMM and inertial ADMM.

## D  Preparatory materials

### D.1  Polynomial extrapolation

Minimal polynomial extrapolation (MPE) [8]: Given $\{z_{k-j}\}_{j=0}^{q+1}$, let $\{v_{k-j}\}_{j=0}^{q}$ be the difference vectors, where $v_j \overset{\text{def}}{=} z_j - z_{j-1}$. Define $V_k = [v_k \quad \cdots \quad v_{k-q}]$.

1. Let $\{c_j\}_{j=1}^{q} \in \text{argmin}_{c \in \mathbb{R}^q} \|V_{k-1}c - v_k\|$, define $c_0 \overset{\text{def}}{=} 1$ and $\gamma_i = c_i / \sum_{i=0}^{q} c_i$ for $i = 0, \ldots, q$.
2. The extrapolated point is then defined to be $\bar{z}_k \overset{\text{def}}{=} \sum_{i=0}^{q} \gamma_i z_{k-i-1}$.

Reduced rank extrapolation (RRE) [15, 28] is obtained by replacing the first step by

$$\{\gamma_j\}_{j=0}^{q} \in \text{argmin}_{\gamma \in \mathbb{R}^{q+1}} \|V_k \gamma\| \text{ subject to } \sum_i \gamma_i = 1.$$

The motivation for the use of such methods for the acceleration of fixed point sequences $x_{k+1} = \mathcal{F}(z_k)$ come from considering the spectral properties of the linearization around the limit point. In particular, if $z^\star$ is the limit point and $z_{k+1} - z^\star = T(z_k - z^\star)$ where $T \in \mathbb{R}^{d \times d}$ and $q$ is the order of the minimal polynomial of $T$ with respect to $z_{k-q-1} - z^\star$ (i.e. $q$ is the monic polynomial of least degree such that $P(T)(z_{k-q-1} - z^\star) = 0$), then one can show that $\bar{z}_k = z^\star$. We refer to [33, 34, 32] for details on these methods and their acceleration guarantees.

(a) Original image

(b) Observed image

(c) ADMM, PSNR = 26.5448

(d) Inertial ADMM, PSNR = 26.1096

(e) A$^3$DMM $s = 100$, PSNR = 27.0402

(f) A$^3$DMM $s = +\infty$, PSNR = 27.0402

Figure C.3: Comparison of image quality at the 30'th iteration of ADMM, inertial ADMM and A$^3$DMM with two different prediction steps.

## D.2 Angle between subspaces

Let $T_1, T_2$ be two subspaces, and without the loss of generality, assume

$$1 \leq p \stackrel{\text{def}}{=} \dim(T_1) \leq q \stackrel{\text{def}}{=} \dim(T_2) \leq n - 1.$$

**Definition D.1 (Principal angles).** The principal angles $\theta_k \in [0, \frac{\pi}{2}]$, $k = 1, \ldots, p$ between sub-spaces $T_1$ and $T_2$ are defined by, with $u_0 = v_0 \overset{\text{def}}{=} 0$, and

$$\cos(\theta_k) \overset{\text{def}}{=} \langle u_k, v_k \rangle = \max \langle u, v \rangle \text{ s.t. } u \in T_1, v \in T_2, \|u\| = 1, \|v\| = 1,$$
$$\langle u, u_i \rangle = \langle v, v_i \rangle = 0, i = 0, \cdots, k-1.$$

The principal angles $\theta_k$ are unique and satisfy $0 \leq \theta_1 \leq \theta_2 \leq \cdots \leq \theta_p \leq \pi/2$.

**Definition D.2 (Friedrichs angle).** The Friedrichs angle $\theta_F \in ]0, \frac{\pi}{2}]$ between $T_1$ and $T_2$ is

$$\cos\left(\theta_F(T_1, T_2)\right) \overset{\text{def}}{=} \max \langle u, v \rangle \text{ s.t. } u \in T_1 \cap (T_1 \cap T_2)^\perp, \|u\| = 1, v \in T_2 \cap (T_1 \cap T_2)^\perp, \|v\| = 1.$$

The following lemma shows the relation between the Friedrichs and principal angles, whose proof can be found in [3, Proposition 3.3].

**Lemma D.3 (Principal angles and Friedrichs angle).** *The Friedrichs angle is exactly $\theta_{d+1}$ where $d \overset{\text{def}}{=} \dim(T_1 \cap T_2)$. Moreover, $\theta_F(T_1, T_2) > 0$.*

### D.3 Riemannian Geometry

Let $\mathcal{M}$ be a $C^2$-smooth embedded submanifold of $\mathbb{R}^n$ around a point $x$. With some abuse of terminology, we shall state $C^2$-manifold instead of $C^2$-smooth embedded submanifold of $\mathbb{R}^n$. The natural embedding of a submanifold $\mathcal{M}$ into $\mathbb{R}^n$ permits to define a Riemannian structure and to introduce geodesics on $\mathcal{M}$, and we simply say $\mathcal{M}$ is a Riemannian manifold. We denote respectively $\mathcal{T}_{\mathcal{M}}(x)$ and $\mathcal{N}_{\mathcal{M}}(x)$ the tangent and normal space of $\mathcal{M}$ at point near $x$ in $\mathcal{M}$.

**Exponential map** Geodesics generalize the concept of straight lines in $\mathbb{R}^n$, preserving the zero acceleration characteristic, to manifolds. Roughly speaking, a geodesic is locally the shortest path between two points on $\mathcal{M}$. We denote by $\mathfrak{g}(t; x, h)$ the value at $t \in \mathbb{R}$ of the geodesic starting at $\mathfrak{g}(0; x, h) = x \in \mathcal{M}$ with velocity $\dot{\mathfrak{g}}(t; x, h) = \frac{d\mathfrak{g}}{dt}(t; x, h) = h \in \mathcal{T}_{\mathcal{M}}(x)$ (which is uniquely defined). For every $h \in \mathcal{T}_{\mathcal{M}}(x)$, there exists an interval $I$ around 0 and a unique geodesic $\mathfrak{g}(t; x, h) : I \to \mathcal{M}$ such that $\mathfrak{g}(0; x, h) = x$ and $\dot{\mathfrak{g}}(0; x, h) = h$. The mapping

$$\mathrm{Exp}_x : \mathcal{T}_{\mathcal{M}}(x) \to \mathcal{M}, \ h \mapsto \mathrm{Exp}_x(h) = \mathfrak{g}(1; x, h),$$

is called *Exponential map*. Given $x, x' \in \mathcal{M}$, the direction $h \in \mathcal{T}_{\mathcal{M}}(x)$ we are interested in is such that

$$\mathrm{Exp}_x(h) = x' = \mathfrak{g}(1; x, h).$$

**Parallel translation** Given two points $x, x' \in \mathcal{M}$, let $\mathcal{T}_{\mathcal{M}}(x), \mathcal{T}_{\mathcal{M}}(x')$ be their corresponding tangent spaces. Define

$$\tau : \mathcal{T}_{\mathcal{M}}(x) \to \mathcal{T}_{\mathcal{M}}(x'),$$

the parallel translation along the unique geodesic joining $x$ to $x'$, which is isomorphism and isometry w.r.t. the Riemannian metric.

**Riemannian gradient and Hessian** For a vector $v \in \mathcal{N}_{\mathcal{M}}(x)$, the Weingarten map of $\mathcal{M}$ at $x$ is the operator $\mathfrak{W}_x(\cdot, v) : \mathcal{T}_{\mathcal{M}}(x) \to \mathcal{T}_{\mathcal{M}}(x)$ defined by

$$\mathfrak{W}_x(\cdot, v) = -\mathcal{P}_{\mathcal{T}_{\mathcal{M}}(x)} dV[h],$$

where $V$ is any local extension of $v$ to a normal vector field on $\mathcal{M}$. The definition is independent of the choice of the extension $V$, and $\mathfrak{W}_x(\cdot, v)$ is a symmetric linear operator which is closely tied to the second fundamental form of $\mathcal{M}$, see [11, Proposition II.2.1].

Let $G$ be a real-valued function which is $C^2$ along the $\mathcal{M}$ around $x$. The covariant gradient of $G$ at $x' \in \mathcal{M}$ is the vector $\nabla_{\mathcal{M}} G(x') \in \mathcal{T}_{\mathcal{M}}(x')$ defined by

$$\langle \nabla_{\mathcal{M}} G(x'), h \rangle = \frac{d}{dt} G\left(\mathcal{P}_{\mathcal{M}}(x' + th)\right)\big|_{t=0}, \ \forall h \in \mathcal{T}_{\mathcal{M}}(x'),$$

where $\mathcal{P}_{\mathcal{M}}$ is the projection operator onto $\mathcal{M}$. The covariant Hessian of $G$ at $x'$ is the symmetric linear mapping $\nabla^2_{\mathcal{M}} G(x')$ from $\mathcal{T}_{\mathcal{M}}(x')$ to itself which is defined as

$$\langle \nabla^2_{\mathcal{M}} G(x')h, h \rangle = \frac{d^2}{dt^2} G\left(\mathcal{P}_{\mathcal{M}}(x' + th)\right)\big|_{t=0}, \ \forall h \in \mathcal{T}_{\mathcal{M}}(x'). \tag{D.1}$$

This definition agrees with the usual definition using geodesics or connections [29]. Now assume that $\mathcal{M}$ is a Riemannian embedded submanifold of $\mathbb{R}^n$, and that a function $G$ has a $C^2$-smooth restriction on $\mathcal{M}$. This can be characterized by the existence of a $C^2$-smooth extension (representative) of $G$, *i.e.* a $C^2$-smooth function $\widetilde{G}$ on $\mathbb{R}^n$ such that $\widetilde{G}$ agrees with $G$ on $\mathcal{M}$. Thus, the Riemannian gradient $\nabla_{\mathcal{M}}G(x')$ is also given by

$$\nabla_{\mathcal{M}}G(x') = \mathcal{P}_{\mathcal{T}_{\mathcal{M}}(x')}\nabla\widetilde{G}(x'), \tag{D.2}$$

and $\forall h \in \mathcal{T}_{\mathcal{M}}(x')$, the Riemannian Hessian reads

$$\begin{aligned}
\nabla_{\mathcal{M}}^2 G(x')h &= \mathcal{P}_{\mathcal{T}_{\mathcal{M}}(x')}\mathrm{d}(\nabla_{\mathcal{M}}G)(x')[h] = \mathcal{P}_{\mathcal{T}_{\mathcal{M}}(x')}\mathrm{d}\big(x' \mapsto \mathcal{P}_{\mathcal{T}_{\mathcal{M}}(x')}\nabla_{\mathcal{M}}\widetilde{G}\big)[h] \\
&= \mathcal{P}_{\mathcal{T}_{\mathcal{M}}(x')}\nabla^2\widetilde{G}(x')h + \mathfrak{W}_{x'}\big(h, \mathcal{P}_{\mathcal{N}_{\mathcal{M}}(x')}\nabla\widetilde{G}(x')\big),
\end{aligned} \tag{D.3}$$

where the last equality comes from [1, Theorem 1]. When $\mathcal{M}$ is an affine or linear subspace of $\mathbb{R}^n$, then obviously $\mathcal{M} = x + \mathcal{T}_{\mathcal{M}}(x)$, and $\mathfrak{W}_{x'}(h, \mathcal{P}_{\mathcal{N}_{\mathcal{M}}(x')}\nabla\widetilde{G}(x')) = 0$, hence (D.3) reduces to

$$\nabla_{\mathcal{M}}^2 G(x') = \mathcal{P}_{\mathcal{T}_{\mathcal{M}}(x')}\nabla^2\widetilde{G}(x')\mathcal{P}_{\mathcal{T}_{\mathcal{M}}(x')}.$$

See [23, 11] for more materials on differential and Riemannian manifolds.

## D.4 Preparatory lemmas

The following lemmas characterize the parallel translation and the Riemannian Hessian of nearby points in $\mathcal{M}$.

**Lemma D.4 ([25, Lemma 5.1]).** *Let $\mathcal{M}$ be a $C^2$-smooth manifold around $x$. Then for any $x' \in \mathcal{M} \cap \mathcal{N}$, where $\mathcal{N}$ is a neighborhood of $x$, the projection operator $\mathcal{P}_{\mathcal{M}}(x')$ is uniquely valued and $C^1$ around $x$, and thus*

$$x' - x = \mathcal{P}_{\mathcal{T}_{\mathcal{M}}(x)}(x' - x) + o(\|x' - x\|).$$

*If moreover $\mathcal{M} = x + \mathcal{T}_{\mathcal{M}}(x)$ is an affine subspace, then $x' - x = \mathcal{P}_{\mathcal{T}_{\mathcal{M}}(x)}(x' - x)$.*

**Lemma D.5 ([26, Lemma B.1]).** *Let $x \in \mathcal{M}$, and $x_k$ a sequence converging to $x$ in $\mathcal{M}$. Denote $\tau_k : \mathcal{T}_{\mathcal{M}}(x_k) \to \mathcal{T}_{\mathcal{M}}(x)$ be the parallel translation along the unique geodesic joining $x$ to $x_k$. Then, for any bounded vector $u \in \mathbb{R}^n$, we have*

$$(\tau_k \mathcal{P}_{\mathcal{T}_{\mathcal{M}}(x_k)} - \mathcal{P}_{\mathcal{T}_{\mathcal{M}}(x)})u = o(\|u\|).$$

The Riemannian gradient and Hessian of partly smooth functions are covered by the lemma below.

**Lemma D.6 ([26, Lemma B.2]).** *Let $x, x'$ be two close points in $\mathcal{M}$, denote $\tau : \mathcal{T}_{\mathcal{M}}(x') \to \mathcal{T}_{\mathcal{M}}(x)$ the parallel translation along the unique geodesic joining $x$ to $x'$. The Riemannian Taylor expansion of $R \in C^2(\mathcal{M})$ around $x$ reads,*

$$\tau\nabla_{\mathcal{M}}R(x') = \nabla_{\mathcal{M}}R(x) + \nabla_{\mathcal{M}}^2 R(x)\mathcal{P}_{\mathcal{T}_{\mathcal{M}}(x)}(x' - x) + o(\|x' - x\|). \tag{D.4}$$

**Lemma D.7 (Riemannian gradient and Hessian).** *If $R \in \mathrm{PSF}_x(\mathcal{M}_x)$, then for any point $x' \in \mathcal{M}_x$ near $x$*

$$\nabla_{\mathcal{M}_x}R(x') = \mathcal{P}_{T_{x'}}(\partial R(x')),$$

*and this does not depend on the smooth representation of $R$ on $\mathcal{M}_x$. In turn, for all $h \in T_{x'}$, let $\widetilde{R}$ be a smooth representative of $R$ on $\mathcal{M}_x$,*

$$\nabla_{\mathcal{M}_x}^2 R(x')h = \mathcal{P}_{T_{x'}}\nabla^2\widetilde{R}(x')h + \mathfrak{W}_{x'}\big(h, \mathcal{P}_{T_{x'}^\perp}\nabla\widetilde{R}(x')\big),$$

*where $\mathfrak{W}_x(\cdot, \cdot) : T_x \times T_x^\perp \to T_x$ is the Weingarten map of $\mathcal{M}_x$ at $x$.*

## D.5 Linearization of proximal mapping

In this part, we present one fundamental result led by partial smoothness, the linearization of proximal mapping. We first discuss the property of the Riemannian Hessian of a partly smooth function. Let $R \in \Gamma_0(\mathbb{R}^n)$ be partly smooth at $\bar{x}$ relative to $\mathcal{M}_{\bar{x}}$ and $\bar{u} \in \partial R(\bar{x})$, define the following smooth perturbation of $R$

$$\overline{R}(x) \stackrel{\text{def}}{=} R(x) - \langle x, \bar{u} \rangle,$$

whose Riemannian Hessian at $\bar{x}$ reads $H_{\overline{R}} \stackrel{\text{def}}{=} \mathcal{P}_{T_{\bar{x}}}\nabla_{\mathcal{M}_{\bar{x}}}^2 \overline{R}(\bar{x})\mathcal{P}_{T_{\bar{x}}}$.

**Lemma D.8 ([26, Lemma 4.2]).** *Let $R \in \Gamma_0(\mathbb{R}^n)$ be partly smooth at $\bar{x}$ relative to $\mathcal{M}_{\bar{x}}$, then $H_{\bar{R}}$ is symmetric positive semi-definite if either of the following is true:*

- *$\bar{u} \in \mathrm{ri}(\partial R(\bar{x}))$ is non-degenerate.*
- *$\mathcal{M}_{\bar{x}}$ is an affine subspace.*

*In turn, $\mathrm{Id} + H_{\bar{R}}$ is invertible and $(\mathrm{Id} + H_{\bar{R}})^{-1}$ is symmetric positive definite with all eigenvalues in $]0, 1]$.*

One consequence of Lemma D.8 is that, we can linearize the generalized proximal mapping. For the sake of generality, let $\gamma > 0$, $R \in \Gamma_0(\mathbb{R}^n)$ and $A \in \mathbb{R}^{p \times n}$, define the following generalized proximal mapping

$$\mathrm{prox}_{\gamma R}^A(\cdot) \stackrel{\text{def}}{=} \mathrm{argmin}_{x \in \mathbb{R}^n} \gamma R(x) + \frac{1}{2}\|Ax - \cdot\|^2.$$

Clearly, $\mathrm{prox}_{\gamma R}^A$ is a single-valued mapping when $A$ has full column rank. Denote $A_{T_{\bar{x}}} \stackrel{\text{def}}{=} A \circ \mathcal{P}_{T_{\bar{x}}}$, it is immediate that $A_{T_{\bar{x}}}^T A_{T_{\bar{x}}}$ is positive semidefinite and invertible along $T_{\bar{x}}$. In the following we denote $(A_{T_{\bar{x}}}^T A_{T_{\bar{x}}})^{-1}$ the inverse along $T_{\bar{x}}$ Denote

$$M_{\bar{R}} = A_{T_{\bar{x}}}(\mathrm{Id} + (A_{T_{\bar{x}}}^T A_{T_{\bar{x}}})^{-1} H_{\bar{R}})^{-1}(A_{T_{\bar{x}}}^T A_{T_{\bar{x}}})^{-1} A_{T_{\bar{x}}}^T.$$

**Lemma D.9.** *Let function $R \in \Gamma_0(\mathbb{R}^n)$ be partly smooth at the point $\bar{x}$ relative to the manifold $\mathcal{M}_{\bar{x}}$ and $\bar{u} \in \mathrm{ri}(\partial R(\bar{x}))$. Suppose that there exists $\gamma > 0$, full column rank $A \in \mathbb{R}^{p \times n}$ and $\bar{w} \in \mathbb{R}^p$ such that $\bar{x} = \mathrm{prox}_{\gamma R}^A(\bar{w})$ and $\bar{u} = -A^T(A\bar{x} - \bar{w})/\gamma$. Let $\{w_k\}_{k \in \mathbb{N}}$ be a sequence such that $w_k \to \bar{w}$ and $x_k = \mathrm{prox}_{\gamma R}^A(w_k) \to \bar{x}$, then for all $k$ large enough, there hold $x_k \in \mathcal{M}_{\bar{x}}$ and*

$$A_{T_{\bar{x}}}(x_k - x_{k-1}) = M_{\bar{R}}(w_k - w_{k-1}) + o(\|w_k - w_{k-1}\|). \tag{D.5}$$

**Remark D.10.** When $A = \mathrm{Id}$, then $\mathrm{prox}_{\gamma R}^A$ reduces to the standard proximal mapping, and (D.5) simplifies to

$$x_k - x_{k-1} = \mathcal{P}_{T_{\bar{x}}}\big(\mathrm{Id} + H_{\bar{R}}\big)^{-1}\mathcal{P}_{T_{\bar{x}}}(w_k - w_{k-1}) + o(\|w_k - w_{k-1}\|).$$

In [24] and references therein, to study the local linear convergence of first-order methods, linearization with respect to the limiting points is provided, that is

$$x_k - \bar{x} = \mathcal{P}_{T_{\bar{x}}}\big(\mathrm{Id} + H_{\bar{R}}\big)^{-1}\mathcal{P}_{T_{\bar{x}}}(w_k - \bar{w}) + o(\|w_k - \bar{w}\|).$$

**Proof.** Since $R$ is proper convex and lower semi-continuous, we have $R(x_k) \to R(\bar{x})$ and $\partial R(x_k) \ni u_k = -A^T(Ax_k - w_k)/\gamma \to \bar{u} \in \mathrm{ri}(\partial R(\bar{x}))$, hence $\mathrm{dist}(u_k, \partial R(\bar{x})) \to 0$. As a result, we have $x_k \in \mathcal{M}_{\bar{x}}$ owing to [21, Theorem 5.3] and $u_k \in \mathrm{ri}(\partial R(x_k))$ owing to [35] for all $k$ large enough.

Denote $T_{x_k}, T_{x_{k-1}}$ the tangent spaces of $\mathcal{M}_{\bar{x}}$ at $x_k$ and $x_{k-1}$. Denote $\tau_k : T_{x_k} \to T_{x_{k-1}}$ the parallel translation along the unique geodesic on $\mathcal{M}_{\bar{x}}$ joining $x_k$ to $x_{k-1}$. From the definition of $x_k$, let $h_k = \gamma u_k$, we get

$$h_k \stackrel{\text{def}}{=} -A^T(Ax_k - w_k) \in \gamma \partial R(x_k) \quad \text{and} \quad h_{k-1} \stackrel{\text{def}}{=} -A^T(Ax_{k-1} - w_{k-1}) \in \gamma \partial R(x_{k-1}).$$

Projecting onto corresponding tangent spaces, applying Lemma D.7 and the parallel translation $\tau_k$ leads to

$$\gamma \tau_k \nabla_{\mathcal{M}_{\bar{x}}} R(x_k) = \tau_k \mathcal{P}_{T_{x_k}}(h_k) = \mathcal{P}_{T_{x_{k-1}}}(h_k) + \big(\tau_k \mathcal{P}_{T_{x_k}} - \mathcal{P}_{T_{x_{k-1}}}\big)(h_k),$$

$$\gamma \nabla_{\mathcal{M}_{\bar{x}}} R(x_{k-1}) = \mathcal{P}_{T_{x_{k-1}}}(h_{k-1}).$$

The difference of the above two equalities yields

$$\begin{aligned}
&\gamma \tau_k \nabla_{\mathcal{M}_{\bar{x}}} R(x_k) - \gamma \nabla_{\mathcal{M}_{\bar{x}}} R(x_{k-1}) - \big(\tau_k \mathcal{P}_{T_{x_k}} - \mathcal{P}_{T_{x_{k-1}}}\big)(h_{k-1}) \\
&= \mathcal{P}_{T_{x_{k-1}}}(h_k - h_{k-1}) + \big(\tau_k \mathcal{P}_{T_{x_k}} - \mathcal{P}_{T_{x_{k-1}}}\big)(h_k - h_{k-1}).
\end{aligned} \tag{D.6}$$

Owing to the monotonicity of sub-differential, *i.e.* $\langle h_k - h_{k-1}, x_k - x_{k-1}\rangle \geq 0$, we get

$$\langle A^T A(x_k - x_{k-1}), x_k - x_{k-1}\rangle \leq \langle A^T(w_k - w_{k-1}), x_k - x_{k-1}\rangle \leq \|A\|\|w_k - w_{k-1}\|\|x_k - x_{k-1}\|.$$

Since $A$ has full column rank, $A^T A$ is symmetric positive definite, and there exists $\kappa > 0$ such that $\kappa \|x_k - x_{k-1}\|^2 \leq \langle A^T A(x_k - x_{k-1}), \, x_k - x_{k-1} \rangle$. Back to the above inequality, we get $\|x_k - x_{k-1}\| \leq \frac{\|A\|}{\kappa} \|w_k - w_{k-1}\|$. Therefore for $\|h_k - h_{k-1}\|$, we get

$$\|h_k - h_{k-1}\| = \|A^T(Ax_k - w_k) - A^T(Ax_{k-1} - w_{k-1})\| \leq \|A\|^2 \|x_k - x_{k-1}\| + \|A\| \|w_k - w_{k-1}\|$$
$$\leq \left( \frac{\|A\|^3}{\kappa} + \|A\| \right) \|w_k - w_{k-1}\|.$$

As a result, owing to Lemma D.5, we have for the term $(\tau_k \mathcal{P}_{T_{x_k}} - \mathcal{P}_{T_{x_{k-1}}})(h_k - h_{k-1})$ in (D.6) that

$$\left( \tau_k \mathcal{P}_{T_{x_k}} - \mathcal{P}_{T_{x_{k-1}}} \right)(h_k - h_{k-1}) = o(\|h_k - h_{k-1}\|) = o(\|w_k - w_{k-1}\|).$$

Define $\overline{R}_{k-1}(x) \stackrel{\text{def}}{=} \gamma R(x) - \langle x, \, h_{k-1} \rangle$ and $H_{\overline{R},k-1} \stackrel{\text{def}}{=} \mathcal{P}_{T_{x_{k-1}}} \nabla^2_{\mathcal{M}_{\bar{x}}} \overline{R}_{k-1}(x_{k-1}) \mathcal{P}_{T_{x_{k-1}}}$, then with Lemma D.6 the Riemannian Taylor expansion, we have for the first line of (D.6)

$$\gamma \tau_k \nabla_{\mathcal{M}_{\bar{x}}} R(x_k) - \gamma \nabla_{\mathcal{M}_{\bar{x}}} R(x_{k-1}) - \left( \tau_k \mathcal{P}_{T_{x_k}} - \mathcal{P}_{T_{x_{k-1}}} \right)(h_{k-1})$$
$$= \tau_k \left( \gamma \nabla_{\mathcal{M}_{\bar{x}}} R(x_k) - \mathcal{P}_{T_{x_k}}(h_{k-1}) \right) - \left( \gamma \nabla_{\mathcal{M}_{\bar{x}}} R(x_{k-1}) - \mathcal{P}_{T_{x_{k-1}}}(h_{k-1}) \right)$$
$$= \tau_k \nabla_{\mathcal{M}_{\bar{x}}} \overline{R}_{k-1}(x_k) - \nabla_{\mathcal{M}_{\bar{x}}} \overline{R}_{k-1}(x_{k-1}) \tag{D.7}$$
$$= H_{\overline{R},k-1}(x_k - x_{k-1}) + o(\|x_k - x_{k-1}\|)$$
$$= H_{\overline{R},k-1}(x_k - x_{k-1}) + o(\|w_k - w_{k-1}\|).$$

Back to (D.6), we get

$$H_{\overline{R},k-1}(x_k - x_{k-1}) = \mathcal{P}_{T_{x_{k-1}}}(h_k - h_{k-1}) + o(\|w_k - w_{k-1}\|). \tag{D.8}$$

Define $\overline{R}(x) \stackrel{\text{def}}{=} \gamma R(x) - \langle x, \, \bar{h} \rangle$ and $H_{\overline{R}} = \mathcal{P}_{T_{\bar{x}}} \nabla^2_{\mathcal{M}_{\bar{x}}} \overline{R}(\bar{x}) \mathcal{P}_{T_{\bar{x}}}$, then from (D.8) that

$$H_{\overline{R}}(x_k - x_{k-1}) + \left( H_{\overline{R},k-1} - H_{\overline{R}} \right)(x_k - x_{k-1})$$
$$= \mathcal{P}_{T_{\bar{x}}}(h_k - h_{k-1}) + \left( \mathcal{P}_{T_{x_{k-1}}} - \mathcal{P}_{T_{\bar{x}}} \right)(h_k - h_{k-1}) + o(\|w_k - w_{k-1}\|). \tag{D.9}$$

Owing to continuity, we have $H_{\overline{R},k-1} \to H_{\overline{R}}$ and $\mathcal{P}_{T_{x_{k-1}}} \to \mathcal{P}_{T_{\bar{x}}}$,

$$\lim_{k \to +\infty} \frac{\|(H_{\overline{R},k-1} - H_{\overline{R}})(x_k - x_{k-1})\|}{\|x_k - x_{k-1}\|} \leq \lim_{k \to +\infty} \frac{\|H_{\overline{R},k-1} - H_{\overline{R}}\| \|x_k - x_{k-1}\|}{\|x_k - x_{k-1}\|} = \lim_{k \to +\infty} \|H_{\overline{R},k-1} - H_{\overline{R}}\| = 0,$$
$$\lim_{k \to +\infty} \frac{\|(\mathcal{P}_{T_{x_{k-1}}} - \mathcal{P}_{T_{\bar{x}}})(w_k - w_{k-1})\|}{\|w_k - w_{k-1}\|} \leq \lim_{k \to +\infty} \frac{\|\mathcal{P}_{T_{x_{k-1}}} - \mathcal{P}_{T_{\bar{x}}}\| \|w_k - w_{k-1}\|}{\|w_k - w_{k-1}\|} = \lim_{k \to +\infty} \|\mathcal{P}_{T_{x_{k-1}}} - \mathcal{P}_{T_{\bar{x}}}\| = 0,$$

and $\lim_{k \to +\infty} \frac{\|(\mathcal{P}_{T_{x_{k-1}}} - \mathcal{P}_{T_{\bar{x}}})(x_k - x_{k-1})\|}{\|x_k - x_{k-1}\|} = 0$. Combining this with the definition of $u_k$, the fact that $x_k - x_{k-1} = \mathcal{P}_{T_{\bar{x}}}(x_k - x_{k-1}) + o(\|x_k - x_{k-1}\|)$ from Lemma D.4, and denoting $A_{T_{\bar{x}}} = A \circ \mathcal{P}_{T_{\bar{x}}}$, equation (D.9) can be written as

$$H_{\overline{R}}(x_k - x_{k-1}) = \mathcal{P}_{T_{\bar{x}}}(u_k - u_{k-1}) + o(\|w_k - w_{k-1}\|)$$
$$= -\mathcal{P}_{T_{\bar{x}}}\left( A^T(Ax_k - w_k) - A^T(Ax_{k-1} - w_{k-1}) \right) + o(\|w_k - w_{k-1}\|)$$
$$= -\mathcal{P}_{T_{\bar{x}}} A^T A(x_k - x_{k-1}) + \mathcal{P}_{T_{\bar{x}}} A^T(w_k - w_{k-1}) + o(\|w_k - w_{k-1}\|) \tag{D.10}$$
$$= -A_{T_{\bar{x}}}^T A_{T_{\bar{x}}}(x_k - x_{k-1}) + A_{T_{\bar{x}}}^T(w_k - w_{k-1}) + o(\|w_k - w_{k-1}\|).$$

Since $A$ has full rank, so is $A_{T_{\bar{x}}}$. Hence $A_{T_{\bar{x}}}^T A_{T_{\bar{x}}}$ is invertible along $T_{\bar{x}}$ and from above we have

$$\left( \mathrm{Id} + (A_{T_{\bar{x}}}^T A_{T_{\bar{x}}})^{-1} H_{\overline{R}} \right)(x_k - x_{k-1}) = (A_{T_{\bar{x}}}^T A_{T_{\bar{x}}})^{-1} A_{T_{\bar{x}}}^T(w_k - w_{k-1}) + o(\|w_k - w_{k-1}\|).$$

Denote $M_{\overline{R}} = A_{T_{\bar{x}}}(\mathrm{Id} + (A_{T_{\bar{x}}}^T A_{T_{\bar{x}}})^{-1} H_{\overline{R}})^{-1}(A_{T_{\bar{x}}}^T A_{T_{\bar{x}}})^{-1} A_{T_{\bar{x}}}^T$, then

$$A_{T_{\bar{x}}}(x_k - x_{k-1}) = M_{\overline{R}}(w_k - w_{k-1}) + o(\|w_k - w_{k-1}\|), \tag{D.11}$$

which concludes the proof. $\qquad\square$

# E Trajectory of ADMM

## E.1 Trajectory of ADMM: both $R, J$ are non-smooth

Given a saddle point $(x^\star, y^\star, \psi^\star)$ of $\mathcal{L}(x, y; \psi)$, the first-order optimality condition entails $-A^T \psi^\star \in \partial R(x^\star)$ and $-B^T \psi^\star \in \partial J(y^\star)$. Below we impose a stronger condition

$$- A^T \psi^\star \in \mathrm{ri}\big(\partial R(x^\star)\big) \quad \text{and} \quad - B^T \psi^\star \in \mathrm{ri}\big(\partial J(y^\star)\big). \tag{ND}$$

Suppose $R \in \mathrm{PSF}_{x^\star}(\mathcal{M}_{x^\star}^R), J \in \mathrm{PSF}_{y^\star}(\mathcal{M}_{y^\star}^J)$ are partly smooth, denote $T_{x^\star}^R, T_{y^\star}^J$ the tangent spaces of $\mathcal{M}_{x^\star}^R, \mathcal{M}_{y^\star}^J$ at $x^\star, y^\star$, respectively. Define the following smooth perturbation of $R, J$,

$$\overline{R}(x) \overset{\text{def}}{=} \tfrac{1}{\gamma}\big(R(x) - \langle x, -A^T \psi^\star \rangle\big), \quad \overline{J}(y) \overset{\text{def}}{=} \tfrac{1}{\gamma}\big(J(y) - \langle w, -B^T \psi^\star \rangle\big), \tag{E.1}$$

their Riemannian Hessian $H_{\overline{R}} \overset{\text{def}}{=} \mathcal{P}_{T_{x^\star}^R} \nabla^2_{\mathcal{M}_{x^\star}^R} \overline{R}(x^\star) \mathcal{P}_{T_{x^\star}^R}$, $H_{\overline{J}} \overset{\text{def}}{=} \mathcal{P}_{T_{x^\star}^J} \nabla^2_{\mathcal{M}_{y^\star}^J} \overline{J}(y^\star) \mathcal{P}_{T_{x^\star}^J}$ and

$$\begin{aligned} M_{\overline{R}} &\overset{\text{def}}{=} A_R\big(\mathrm{Id} + (A_R^T A_R)^{-1} H_{\overline{R}}\big)^{-1}(A_R^T A_R)^{-1} A_R^T, \\ M_{\overline{J}} &\overset{\text{def}}{=} B_J\big(\mathrm{Id} + (B_J^T B_J)^{-1} H_{\overline{J}}\big)^{-1}(B_J^T B_J)^{-1} B_J^T, \end{aligned} \tag{E.2}$$

where $A_R \overset{\text{def}}{=} A \circ \mathcal{P}_{T_{x^\star}^R}, B_J \overset{\text{def}}{=} B \circ \mathcal{P}_{T_{y^\star}^J}$. Finally, define

$$M \overset{\text{def}}{=} \tfrac{1}{2}\mathrm{Id} + \tfrac{1}{2}(2M_{\overline{R}} - \mathrm{Id})(2M_{\overline{J}} - \mathrm{Id}). \tag{E.3}$$

**Proof of Theorem 2.2.** The proof of Theorem 2.2 is split into several steps: finite manifold identification of ADMM, local linearization based on partial smoothness, spectral properties of the linearised matrix, and the trajectory of $\{z_k\}_{k \in \mathbb{N}}$. Let $(x^\star, y^\star, \psi^\star)$ be a saddle-point of $\mathcal{L}(x, y; \psi)$.

**1. Finite manifold identification of ADMM** The finite manifold identification of ADMM is already discussed in [27], below we present a short discussion for the sake of self-consistency. At convergence of ADMM, owing to (2) we have

$$A^T \psi^\star = \gamma A^T \big(Ax^\star - \tfrac{1}{\gamma}(z^\star - 2\psi^\star)\big) \quad \text{and} \quad B^T \psi^\star = \gamma B^T \big(By^\star - \tfrac{1}{\gamma}(z^\star - \gamma b)\big).$$

From the update of $x_k, y_k$ in (2), we have the following monotone inclusions

$$\begin{aligned} -\gamma A^T \big(Ax_k - \tfrac{1}{\gamma}(z_{k-1} - 2\psi_{k-1})\big) &\in \partial R(x_k) \quad \text{and} \quad -\gamma B^T \big(By_k - \tfrac{1}{\gamma}(z_k - \gamma b)\big) \in \partial J(y_k), \\ -\gamma A^T \big(Ax^\star - \tfrac{1}{\gamma}(z^\star - 2\psi^\star)\big) &\in \partial R(x^\star) \quad \text{and} \quad -\gamma B^T \big(By^\star - \tfrac{1}{\gamma}(z^\star - \gamma b)\big) \in \partial J(y^\star). \end{aligned}$$

Since $A$ is bounded, it then follows that

$$\begin{aligned} \mathrm{dist}\big(-A^T \psi^\star, \partial R(x_k)\big) &\le \gamma \|A^T \big(Ax_k - \tfrac{1}{\gamma}(z_{k-1} - 2\psi_{k-1})\big) - A^T \big(Ax^\star - \tfrac{1}{\gamma}(z^\star - 2\psi^\star)\big)\| \\ &\le \gamma \|A\|\|A(x_k - x^\star) - \tfrac{1}{\gamma}(z_{k-1} - z^\star) + \tfrac{2}{\gamma}(\psi_{k-1} - \psi^\star)\| \\ &\le \gamma \|A\|\big(\|A\|\|x_k - x^\star\| + \tfrac{1}{\gamma}\|z_{k-1} - z^\star\| + \tfrac{2}{\gamma}\|\psi_{k-1} - \psi^\star\|\big) \to 0. \end{aligned}$$

and similarly

$$\mathrm{dist}\big(-B^T \psi^\star, \partial J(y_k)\big) \le \gamma \|B\|\big(\|B\|\|y_k - y^\star\| + \tfrac{1}{\gamma}\|z_k - z^\star\|\big) \to 0.$$

Since $R \in \Gamma_0(\mathbb{R}^n)$ and $J \in \Gamma_0(\mathbb{R}^m)$, then by the sub-differentially continuous property of them we have $R(x_k) \to R(x^\star)$ and $J(y_k) \to J(y^\star)$. Hence the conditions of [21, Theorem 5.3] are fulfilled for $R$ and $J$, and there exists $K$ large enough such that for all $k \ge K$, there holds

$$(x_k, y_k) \in \mathcal{M}_{x^\star}^R \times \mathcal{M}_{y^\star}^J,$$

which is the finite manifold identification.

**2. linearization of ADMM** For convenience, denote $\beta = 1/\gamma$. For the update of $y_k$, define $w_k = -\beta(z_k - \gamma b)$, we have from (2) that

$$y_k = \text{argmin}_{y \in \mathbb{R}^m} \beta J(y) + \frac{1}{2}\|By - w_k\|^2.$$

Owing to the optimality condition of a saddle point, define $\bar{J}(y) \overset{\text{def}}{=} \beta J(y) - \langle y, -\beta B^T \psi^\star\rangle$ and its Riemannian Hessian $H_{\bar{J}} = \mathcal{P}_{T_{y^\star}^J} \nabla^2_{\mathcal{M}_{y^\star}^J} \bar{J}(y^\star) \mathcal{P}_{T_{y^\star}^J}$. For $B$, define $B_J = B \circ \mathcal{P}_{T_{y^\star}^J}$, and $M_{\bar{J}} = B_J(\text{Id} + (B_J^T B_J)^{-1} H_{\bar{J}})^{-1}(B_J^T B_J)^{-1}B_J^T$. Then owing to Lemma D.9, we get

$$\begin{aligned} B_J(y_k - y_{k-1}) &= M_{\bar{J}}(w_k - w_{k-1}) + o(\|w_k - w_{k-1}\|) \\ &= -\beta M_{\bar{J}}(z_k - z_{k-1}) + o(\|z_k - z_{k-1}\|). \end{aligned} \tag{E.4}$$

Now consider $x_k$ and let $w_k = \beta(z_{k-1} - 2\psi_{k-1})$, we get from (2) that

$$x_k = \text{argmin}_{x \in \mathbb{R}^n} \beta R(x) + \frac{1}{2}\|Ax - w_k\|^2.$$

Define $\bar{R}(x) \overset{\text{def}}{=} \beta R(x) - \langle x, -\beta A^T \psi^\star\rangle$ and its Riemannian Hessian $H_{\bar{R}} = \mathcal{P}_{T_{x^\star}^R} \nabla^2_{\mathcal{M}_{x^\star}^R} \bar{R}(x^\star) \mathcal{P}_{T_{x^\star}^R}$. Denote $A_R = A \circ \mathcal{P}_{T_{x^\star}^R}$, and $M_{\bar{R}} = A_R(\text{Id} + (A_R^T A_R)^{-1} H_{\bar{R}})^{-1}(A_R^T A_R)^{-1}A_R^T$. Note from (2) that $\psi_{k-1} - \psi_{k-2} = z_{k-1} - z_{k-2} + \gamma B(y_{k-1} - y_{k-2})$, then

$$\begin{aligned} w_k - w_{k-1} &= \beta(z_{k-1} - z_{k-2}) - 2\beta(\psi_{k-1} - \psi_{k-2}) \\ &= -\beta(z_{k-1} - z_{k-2}) - 2\beta\gamma B(y_{k-1} - y_{k-2}) \\ &= -\beta(z_{k-1} - z_{k-2}) - 2B_J(y_{k-1} - y_{k-2}) + o(\|y_{k-1} - y_{k-2}\|), \end{aligned}$$

where $y_{k-1} - y_{k-2} = \mathcal{P}_{T_{x^\star}^R}(y_{k-1} - y_{k-2}) + o(\|y_{k-1} - y_{k-2}\|)$ from Lemma D.4 is applied. From (B.7), we have $o(\|y_{k-1} - y_{k-2}\|) = o(\|z_{k-1} - z_{k-2}\|)$ and $o(\|w_{k-1} - w_{k-2}\|) = o(\|z_{k-1} - z_{k-2}\|)$, then applying Lemma D.9 yields,

$$\begin{aligned} A_R(x_k - x_{k-1}) &= M_{\bar{R}}(w_k - w_{k-1}) + o(\|w_k - w_{k-1}\|) \\ &= -\beta M_{\bar{R}}(z_{k-1} - z_{k-2}) + 2M_{\bar{R}}B_J(y_{k-1} - y_{k-2}) + o(\|z_{k-1} - z_{k-2}\|) \\ &= -\beta M_{\bar{R}}(z_{k-1} - z_{k-2}) + 2\beta M_{\bar{R}}M_{\bar{J}}(z_{k-1} - z_{k-2}) + o(\|z_{k-1} - z_{k-2}\|). \end{aligned} \tag{E.5}$$

Finally, from (2), (E.4) and (E.5), we have that

$$\begin{aligned} z_k - z_{k-1} &= \big(z_{k-1} + \gamma(Ax_k + By_{k-1} - b)\big) - \big(z_{k-2} + \gamma(Ax_{k-1} + By_{k-2} - b)\big) \\ &= (z_{k-1} - z_{k-2}) + \gamma A(x_k - x_{k-1}) + \gamma B(y_{k-1} - y_{k-2}) \\ &= (z_{k-1} - z_{k-2}) + \gamma A_R(x_k - x_{k-1}) + \gamma B_J(y_{k-1} - y_{k-2}) + o(\|z_{k-1} - z_{k-2}\|) \\ &= (z_{k-1} - z_{k-2}) - M_{\bar{R}}(z_{k-1} - z_{k-2}) + 2M_{\bar{R}}M_{\bar{J}}(z_{k-1} - z_{k-2}) \\ &\quad + M_{\bar{J}}(z_{k-1} - z_{k-2}) + o(\|z_{k-1} - z_{k-2}\|) \\ &= \big(\text{Id} + 2M_{\bar{R}}M_{\bar{J}} - M_{\bar{R}} - M_{\bar{J}}\big)(z_{k-1} - z_{k-2}) + o(\|z_{k-1} - z_{k-2}\|), \end{aligned}$$

which is the desired linearization of ADMM.

**3. Spectral properties of $M$** Consider first the case where both $R, J$ are general partly smooth functions, under which we can shown the non-expansiveness of $M$. For $M_{\bar{R}}$, since $A$ is injective, so is $A_R$, then $A_R^T A_R$ is symmetric positive definite. Therefore, we have the following similarity result for $M_{\bar{R}}$,

$$\begin{aligned} M_{\bar{R}} &= A_R\big((A_R^T A_R)^{-\frac{1}{2}}\big(\text{Id} + (A_R^T A_R)^{-\frac{1}{2}}H_{\bar{R}}(A_R^T A_R)^{-\frac{1}{2}}\big)(A_R^T A_R)^{\frac{1}{2}}\big)^{-1}(A_R^T A_R)^{-1}A_R^T \\ &= A_R(A_R^T A_R)^{-\frac{1}{2}}\big(\text{Id} + (A_R^T A_R)^{-\frac{1}{2}}H_{\bar{R}}(A_R^T A_R)^{-\frac{1}{2}}\big)^{-1}(A_R^T A_R)^{\frac{1}{2}}(A_R^T A_R)^{-1}A_R^T \\ &= A_R(A_R^T A_R)^{-\frac{1}{2}}\big(\text{Id} + (A_R^T A_R)^{-\frac{1}{2}}H_{\bar{R}}(A_R^T A_R)^{-\frac{1}{2}}\big)^{-1}(A_R^T A_R)^{-\frac{1}{2}}A_R^T. \end{aligned} \tag{E.6}$$

Since $(A_R^T A_R)^{-\frac{1}{2}}H_{\bar{R}}(A_R^T A_R)^{-\frac{1}{2}}$ is symmetric positive definite, hence maximal monotone, then the matrix

$$\big(\text{Id} + (A_R^T A_R)^{-\frac{1}{2}}H_{\bar{R}}(A_R^T A_R)^{-\frac{1}{2}}\big)^{-1}$$

is firmly non-expansive. Let $A_R = USV^T$ be the SVD of $A_R$, then we have

$$\|A_R(A_R^T A_R)^{-\frac{1}{2}}\| = \|USV^T(VSU^T USV^T)^{-\frac{1}{2}}\| = \|USV^T(VS^2V^T)^{-\frac{1}{2}}\| = \|USV^T VS^{-1}V^T\| = 1.$$

Then owing to [2, Example 4.14], $M_{\overline{R}}$ is firmly non-expansive. Similarly, $M_{\overline{J}}$ is firmly non-expansive, and so is $M$ [2, Proposition 4.31]. Therefore, the power $M^k$ is convergent.

Now suppose that both $R, J$ are locally polyhedral around $(x^\star, y^\star)$, then $M_{\overline{R}}$ and $M_{\overline{J}}$ become

$$M_{\overline{R}} = A_R(A_R^T A_R)^{-1}A_R^T \quad \text{and} \quad M_{\overline{J}} = B_J(B_J^T B_J)^{-1}B_J^T,$$

which are projection operators onto the ranges of $A_R$ and $B_J$, respectively. Denote these two subspaces by $T_{A_R}$ and $T_{B_J}$, and correspondingly $\mathcal{P}_{T_{A_R}} \overset{\text{def}}{=} A_R(A_R^T A_R)^{-1}A_R^T$ and $\mathcal{P}_{T_{B_J}} \overset{\text{def}}{=} B_J(B_J^T B_J)^{-1}B_J^T$. Then

$$M = \mathcal{P}_{T_{A_R}}\mathcal{P}_{T_{B_J}} + (\text{Id} - \mathcal{P}_{T_{A_R}})(\text{Id} - \mathcal{P}_{T_{B_J}}).$$

Denote the dimension of $T_{A_R}, T_{B_J}$ by $\dim(T_{A_R}) = p, \dim(T_{B_J}) = q$, and the dimension of the intersection $\dim(T_{A_R} \cap T_{B_J}) = d$. Without the loss of generality, we assume that $1 \leq p \leq q \leq n$. Consequently, there are $r = p - d$ principal angles $(\zeta_i)_{i=1,\ldots,r}$ between $T_{A_R}$ and $T_{B_J}$ that are strictly greater than 0 and smaller than $\pi/2$. Suppose that $\zeta_1 \leq \cdots \leq \zeta_r$. Define the following two diagonal matrices

$$C = \text{diag}\big(\cos(\zeta_1), \cdots, \cos(\zeta_r)\big) \quad \text{and} \quad S = \text{diag}\big(\sin(\zeta_1), \cdots, \sin(\zeta_r)\big).$$

Owing to [4, 13], there exists a real orthogonal matrix $U$ such that

$$M = U \left[\begin{array}{cc|cc} C^2 & CS & 0 & 0 \\ -CS & C^2 & 0 & 0 \\ \hline 0 & 0 & 0_{q-p+2d} & 0 \\ 0 & 0 & 0 & \text{Id}_{n-p-q} \end{array}\right] U^T,$$

which indicates $M$ is normal and all its eigenvalues are inside unit disc.

Let $M^\infty = \lim_{k\to+\infty} M^k$ and $\widetilde{M} = M - M^\infty$, then we have

$$\widetilde{M} = U \left[\begin{array}{cc|c} C^2 & CS & 0 \\ -CS & C^2 & 0 \\ \hline 0 & 0 & 0_{n-2r} \end{array}\right] U^T. \tag{E.7}$$

**4. Trajectory of ADMM** Owing to the polyhedrality of $R$ and $J$, all the small $o$-terms in the linearization proof vanish and we get directly

$$z_k - z_{k-1} = M(z_{k-1} - z_{k-2}) = M^k(z_0 - z_{-1}). \tag{E.8}$$

As $v_k \overset{\text{def}}{=} z_k - z_{k-1} \to 0$, passing to the limit we get from above

$$0 = \lim_{k\to+\infty} M^k v_0 = M^\infty v_0,$$

which means $v_0 \in \ker(M)$ where $\ker(M)$ denotes the kernel of $M$. Since $M^\infty M^k = M^\infty$, we have $v_k \in \ker(M)$ holds for any $k \in \mathbb{N}$. Then from (E.8) we have

$$v_k = (M - M^\infty)v_k = \widetilde{M}v_{k-1}.$$

The block diagonal property of (E.7) indicates that there exists an elementary transformation matrix $E$ such that

$$\widetilde{M} = UE \left[\begin{array}{cccc} B_1 & & & \\ & \ddots & & \\ & & B_r & \\ & & & 0_{n-2r} \end{array}\right] EU^T,$$

where for each $i = 1, ..., r$, we have

$$B_i = \cos(\zeta_i) \left[\begin{array}{cc} \cos(\zeta_i) & \sin(\zeta_i) \\ -\sin(\zeta_i) & \cos(\zeta_i) \end{array}\right]$$

which is rotation matrix scaled by $\cos(\zeta_i)$. It is easy to show that, for each $i = 1, ..., d$, there holds

$$\lim_{k \to +\infty} B_i^k = 0,$$

since the spectral radius of $B_i$ is $\rho(B_i) = \cos(\zeta_i) < 1$.

Suppose for some $1 \le e < r$, we have

$$\zeta = \zeta_1 = \cdots = \zeta_e < \zeta_{e+1} \le \cdots \le \zeta_r.$$

Consider the following decompositions

$$\Gamma_1 = \begin{bmatrix} B_1 & & & \\ & \ddots & & \\ & & B_e & \\ & & & 0_{n-2e} \end{bmatrix} \quad \text{and} \quad \Gamma_2 = \begin{bmatrix} B_1 & & & \\ & \ddots & & \\ & & B_r & \\ & & & 0_{n-2r} \end{bmatrix} - \Gamma_1.$$

Denote $\eta = \frac{\cos(\zeta_{e+1})}{\cos(\zeta)}$, it is immediate to see that $\frac{1}{\cos^k(\zeta)} \Gamma_2^k = O(\eta^k) \to 0$, and for each $i = 1, ..., e$

$$\frac{1}{\cos(\zeta)} B_i = \begin{bmatrix} \cos(\zeta) & \sin(\zeta) \\ -\sin(\zeta) & \cos(\zeta) \end{bmatrix}$$

which is a circular rotation. Therefore, $\frac{1}{\cos(\zeta)} \Gamma_1$ is a rotation with respect to the first $2e$ elements. Denote $u_k = EU^T v_k$, then from $v_k = \widetilde{M} v_{k-1} = UE(\Gamma_1 + \Gamma_2)EU^T v_k$, we get

$$u_k = (\Gamma_1 + \Gamma_2)u_k = (\Gamma_1 + \Gamma_2)^k u_0 = \Gamma_1^k u_0 + \Gamma_2^k u_0,$$

which is an orthogonal decomposition of $u_k$. Define

$$s_k = \frac{1}{\cos^k(\zeta)} \Gamma_1^k u_1 \quad \text{and} \quad t_k = \frac{1}{\cos^k(\zeta)} \Gamma_2^k u_1,$$

then we have that $\|s_k\| = \|s_{k-1}\|$ and $\langle s_k, s_{k-1} \rangle = \cos(\zeta)\|s_k\|^2$, and $t_k = O(\eta^k)$. As a result, for $\cos(\theta_k)$ we have

$$\begin{aligned}
\cos(\theta_k) &= \frac{\langle v_k, v_{k-1} \rangle}{\|v_k\| \|v_{k-1}\|} = \frac{\langle u_k, u_{k-1} \rangle}{\|u_k\| \|u_{k-1}\|} = \frac{\langle s_k + t_k, s_{k-1} + t_{k-1} \rangle}{\|s_k + t_k\| \|s_{k-1} + t_{k-1}\|} \\
&= \frac{\langle s_k, s_{k-1} \rangle}{\|s_k + t_k\| \|s_{k-1} + t_{k-1}\|} + \frac{\langle t_k, t_{k-1} \rangle}{\|s_k + t_k\| \|s_{k-1} + t_{k-1}\|} \quad \text{(E.9)} \\
&= \frac{\|s_k\|^2 \cos(\zeta)}{\|s_k\|^2 + \|t_k\|^2} \cdot \frac{\|s_k + t_k\|}{\|s_{k-1} + t_{k-1}\|} + O(\eta^{2k-1}).
\end{aligned}$$

Using the fact that

$$\frac{\|s_k\|^2 \cos(\zeta)}{\|s_k\|^2 + \|t_k\|^2} = \cos(\zeta)\big(1 - \|t_k\|^2 + O(\|t_k\|^4)\big) = \cos(\zeta) + O(\eta^{2k}) \quad \text{and} \quad \frac{\|s_k + t_k\|}{\|s_{k-1} + t_{k-1}\|} \to 1$$

we conclude that $\cos(\theta_k) \to \cos(\zeta)$. As a matter of fact, we have $\cos(\theta_k) - \cos(\zeta) = O(\eta^{2k})$ which shows how fast $\cos(\theta_k)$ converges to $\cos(\zeta)$. $\qquad \square$

### E.2   Trajectory of ADMM: $R$ or/and $J$ is smooth

Now we consider the case that at least one function out of $R, J$ is smooth. For simplicity, consider that $R$ is smooth and $J$ remains non-smooth. Assume that $R$ is locally $C^2$-smooth around $x^\star$, the Hessian of $R$ at $x^\star$ reads $\nabla^2 R(x^\star)$ which is positive semi-definite owing to convexity. Define $M_R \overset{\text{def}}{=} A\big(\text{Id} + \frac{1}{\gamma}(A^T A)^{-1}\nabla^2 R(x^\star)\big)^{-1}(A^T A)^{-1}A^T$, and redefine

$$M \overset{\text{def}}{=} \frac{1}{2}\text{Id} + \frac{1}{2}(2M_R - \text{Id})(2M_{\bar{J}} - \text{Id}). \tag{E.10}$$

**Proof of Proposition 2.4.** We prove the corollary in two steps.

**1. Linearization of ADMM** Following the above proof, we have for $y_k$ that
$$B_J(y_k - y_{k-1}) = \beta M_{\bar{J}}(z_k - z_{k-1}) + o(\|z_k - z_{k-1}\|).$$
From (2), for $x_{k+1}$ and $x_k$, since $R$ is globally smooth differentiable
$$-A^T(Ax_k - \beta(z_{k-1} - 2\psi_{k-1})) \in \beta\nabla R(x_k) \quad \text{and} \quad -A^T(Ax_{k-1} - \beta(z_{k-2} - 2\psi_{k-2})) \in \beta\nabla R(x_{k-1}),$$
which leads to, applying the local $C^2$-smoothness of $R$ around $x^\star$
$$\begin{aligned}
&- A^T(Ax_k - \beta(z_{k-1} - 2\psi_{k-1})) + A^T(Ax_{k-1} - \beta(z_{k-2} - 2\psi_{k-2}))\\
&= \beta\nabla R(x_k) - \beta\nabla R(x_{k-1})\\
&= \beta\nabla^2 R(x_{k-1})(x_k - x_{k-1}) + o(\|x_k - x_{k-1}\|)\\
&= \beta\nabla^2 R(x^\star)(x_k - x_{k-1}) + \beta(\nabla^2 R(x_{k-1}) - \nabla^2 R(x^\star))(x_k - x_{k-1}) + o(\|x_k - x_{k-1}\|)\\
&= \beta\nabla^2 R(x^\star)(x_k - x_{k-1}) + o(\|z_{k-1} - z_{k-2}\|).
\end{aligned}$$
Using the fact that $A^T A$ is invertible and rearranging terms, we arrive at
$$\begin{aligned}
&\big(\mathrm{Id} + \beta(A^T A)^{-1}\nabla^2 R(x^\star)\big)(x_k - x_{k-1}) + o(\|z_{k-1} - z_{k-2}\|)\\
&= \beta(A^T A)^{-1}A^T(z_{k-1} - z_{k-2}) - 2\beta(A^T A)^{-1}A^T(\psi_{k-1} - \psi_{k-2}) + o(\|z_{k-1} - z_{k-2}\|)\\
&= -\beta(A^T A)^{-1}A^T(z_{k-1} - z_{k-2}) + 2(A^T A)^{-1}A^T B_J(y_{k-1} - y_{k-2}) + o(\|z_{k-1} - z_{k-2}\|),
\end{aligned}$$
which further leads to, denote $M_R = A(\mathrm{Id} + (A^T A)^{-1}H_R)^{-1}(A^T A)^{-1}A^T$
$$\begin{aligned}
A(x_k - x_{k-1}) &= -\beta M_R(z_{k-1} - z_{k-2}) + 2M_R B_J(y_{k-1} - y_{k-2}) + o(\|z_{k-1} - z_{k-2}\|)\\
&= -\beta M_R(z_{k-1} - z_{k-2}) + 2\beta M_R M_{\bar{J}}(z_{k-1} - z_{k-2}) + o(\|z_{k-1} - z_{k-2}\|).
\end{aligned}$$
Finally, from (2), we have that
$$z_k - z_{k-1} = \big(\mathrm{Id} + 2M_R M_{\bar{J}} - M_R - M_{\bar{J}}\big)(z_{k-1} - z_{k-2}) + o(\|z_{k-1} - z_{k-2}\|).$$

**2. Trajectory of ADMM** Since $A$ is full rank square matrix and hence invertible, from (E.6) we have
$$\begin{aligned}
M_R &= A(\mathrm{Id} + \tfrac{1}{\gamma}(A^T A)^{-1}\nabla^2 R(x^\star))^{-1}(A^T A)^{-1}A^T\\
&= A(A^T A)^{-\frac{1}{2}}\big(\mathrm{Id} + \tfrac{1}{\gamma}(A^T A)^{-\frac{1}{2}}\nabla^2 R(x^\star)(A^T A)^{-\frac{1}{2}}\big)^{-1}(A^T A)^{-\frac{1}{2}}A^T\\
&\sim \big(\mathrm{Id} + \tfrac{1}{\gamma}(A^T A)^{-\frac{1}{2}}\nabla^2 R(x^\star)(A^T A)^{-\frac{1}{2}}\big)^{-1},
\end{aligned}$$
where $\big(\mathrm{Id} + \tfrac{1}{\gamma}(A^T A)^{-\frac{1}{2}}\nabla^2 R(x^\star)(A^T A)^{-\frac{1}{2}}\big)^{-1}$ is symmetric positive definite. If we choose $\gamma$ such that
$$\tfrac{1}{\gamma}\|(A^T A)^{-\frac{1}{2}}\nabla^2 R(x^\star)(A^T A)^{-\frac{1}{2}}\| < 1,$$
then all the eigenvalues of $M_R$ are in $]1/2, 1]$, hence $W_R \stackrel{\text{def}}{=} 2M_R - \mathrm{Id}$ is symmetric positive definite. Therefore, we get
$$\begin{aligned}
\tfrac{1}{2}\mathrm{Id} + \tfrac{1}{2}W_R(2M_{\bar{J}} - \mathrm{Id}) &= W_R^{1/2}\big(\tfrac{1}{2}\mathrm{Id} + \tfrac{1}{2}W_R^{1/2}(2M_{\bar{J}} - \mathrm{Id})W_R^{1/2}\big)W_R^{-1/2}\\
&\sim \tfrac{1}{2}\mathrm{Id} + \tfrac{1}{2}W_R^{1/2}(2M_{\bar{J}} - \mathrm{Id})W_R^{1/2},
\end{aligned}$$
and $\overline{M} \stackrel{\text{def}}{=} \tfrac{1}{2}\mathrm{Id} + \tfrac{1}{2}W_R^{1/2}(2M_{\bar{J}} - \mathrm{Id})W_R^{1/2}$ is symmetric positive semi-definite with all eigenvalues in $[0, 1]$. Hence, by similarity, the eigenvalues of $M$ are all real and contained in $[0, 1]$. $\qquad\square$

# F  Adaptive acceleration for ADMM

## F.1  Convergence of $A^3$DMM

**Proof of Proposition 4.2.** From the perturbation formulation $z_k = \mathcal{F}(z_{k-1} + \varepsilon_{k-1})$, we have that
$$z_k = \mathcal{F}(z_{k-1} + \varepsilon_{k-1}) = \mathcal{F}(z_{k-1}) + \big(\mathcal{F}(z_{k-1} + \varepsilon_{k-1}) - \mathcal{F}(z_{k-1})\big).$$
Given any $z^\star \in \mathrm{fix}(\mathcal{F})$, since $\mathcal{F}$ is firmly non-expansive, hence non-expansive, we have
$$\|z_k - z^\star\| \le \|\mathcal{F}(z_{k-1}) - \mathcal{F}(z^\star)\| + \|\mathcal{F}(z_{k-1} + \varepsilon_k) - \mathcal{F}(z_{k-1})\| \le \|z_{k-1} - z^\star\| + \|\varepsilon_{k-1}\|,$$
which means that $\{z_k\}_{k\in\mathbb{N}}$ is quasi-Fejér monotone with respect to $\mathrm{fix}(\mathcal{F})$. Then invoke [2, Proposition 5.34] we obtain the convergence of the sequence $\{z_k\}_{k\in\mathbb{N}}$. $\qquad\square$

## F.2 Acceleration guarantee of A³DMM

Recall the definition of $V_{k-1}, c_k, C_k$ and $\bar{z}_{k,s}$ in the beginning of the section. By definition,

$$V_k = MV_{k-1}. \tag{F.1}$$

Define $E_{k,j} \stackrel{\text{def}}{=} V_k C_k^j - V_{k+1}$ for $j \geq 1$ and

$$E_{k,0} \stackrel{\text{def}}{=} V_{k-1}C_k - V_k = [(V_{k-1}c_k - v_k) \quad 0 \quad \cdots \quad 0]. \tag{F.2}$$

We obtain the relation between the extrapolated point $\bar{z}_{k,s}$ and the $(k+s)$'th point of $\{z_k\}_{k\in\mathbb{N}}$

$$\bar{z}_{k,s} = z_k + \sum_{j=1}^{s}(v_{j+k} + (E_{k,j})_{(:,1)}) = z_{k+s} + \sum_{j=1}^{s}(E_{k,j})_{(:,1)}$$

In the following, given a matrix $M$, we let $\rho(M)$ denote the spectral radius of $M$ and $\lambda(M)$ denote its spectrum.

**Proof of Proposition 4.3.** We first prove $(i)$ that the extrapolation error is controlled by the coefficients fitting error. Since $k \in \mathbb{N}$ is fixed, for ease of notation, we also write $E_\ell = E_{k,\ell}$ and $C = C_k$. We first show that for $\ell \in \mathbb{N}$, we have

$$E_\ell = \sum_{j=1}^{\ell} M^j E_0 C^{\ell-j}. \tag{F.3}$$

We prove this by induction. Note that

$$V_k C \stackrel{\text{(F.1)}}{=} (MV_{k-1})C \stackrel{\text{(F.2)}}{=} MV_k + ME_0 \stackrel{\text{(F.1)}}{=} V_{k+1} + ME_0.$$

Therefore, $E_1 = ME_0$ as required. Assume that (F.4) is true up to $\ell = m$. Then,

$$V_k C^{m+1} \stackrel{\text{(F.1)}}{=} (MV_{k-1})C^{m+1} \stackrel{\text{(F.2)}}{=} MV_k C^m + ME_0 C^m = M(V_{m+k} + E_m) + ME_0 C^m$$

$$\stackrel{\text{(F.1)}}{=} V_{m+2} + ME_m + ME_0 C^m$$

So, plugging in our assumption on $E_m$, we have

$$E_{m+1} = ME_m + ME_0 C^m = ME_0 C^m + M\left(\sum_{j=1}^{m} M^j E_0 C^{m-j}\right) = \sum_{j=1}^{m+1} M^j E_0 C^{m+1-j}.$$

To bound the extrapolation error,

$$\sum_{m=1}^{s} E_m = \sum_{m=1}^{s}\left(\sum_{j=1}^{m} M^j E_0 C^{m-j}\right) = \sum_{\ell=0}^{s-1}\left(\sum_{j=1}^{s-\ell} M^j\right)E_0 C^\ell = \sum_{\ell=1}^{s} M^\ell E_0 \left(\sum_{i=0}^{s-\ell} C^i\right)$$

Therefore,

$$\|\bar{z}_{k,s} - z^\star\| \leq \|z_{k+s} - z^\star\| + \sum_{\ell=1}^{s}\|M^\ell\|\|E_0\|\left\|\sum_{i=0}^{s-\ell} C^i_{(1,1)}\right\|.$$

In the case of $s = +\infty$, we have

$$\|\bar{z}_{k,\infty} - z^\star\| \leq \sum_{\ell=1}^{\infty}\|M^\ell\|\|E_0(\text{Id} - C)^{-1}_{(:,1)}\| = \frac{\|E_0\|}{1 - \sum_i c_i}\sum_{\ell=1}^{\infty}\|M^\ell\|.$$

The fact that $B_s$ is uniformly bounded in $s$ if $\rho(M) < 1$ and $\rho(C) < 1$ follows because this implies that $\sum_{\ell=1}^{\infty}\|M^\ell\| < \infty$ thanks to the Gelfand formula, and $\sum_{i=0}^{\infty} C^i = (\text{Id} - C)^{-1}$ and its $(1,1)^{th}$ entry is precisely $\frac{1}{1-\sum_i c_i}$. Since $k \in \mathbb{N}$ is fixed, for ease of notation, we also write $E_\ell = E_{k,\ell}$ and $C = C_k$. We first show that for $\ell \in \mathbb{N}$, we have

$$E_\ell = \sum_{j=1}^{\ell} M^j E_0 C^{\ell-j}. \tag{F.4}$$

We prove this by induction. Note that

$$V_k C \stackrel{\text{(F.1)}}{=} (MV_{k-1})C \stackrel{\text{(F.2)}}{=} MV_k + ME_0 \stackrel{\text{(F.1)}}{=} V_{k+1} + ME_0.$$

Therefore, $E_1 = ME_0$ as required. Assume that (F.4) is true up to $\ell = m$. Then,

$$V_k C^{m+1} \stackrel{\text{(F.1)}}{=} (MV_{k-1})C^{m+1}$$

$$\stackrel{\text{(F.2)}}{=} MV_k C^m + ME_0 C^m = M(V_{m+k} + E_m) + ME_0 C^m$$

$$\stackrel{\text{(F.1)}}{=} V_{m+2} + ME_m + ME_0 C^m.$$

So, plugging in our assumption on $E_m$, we have

$$E_{m+1} = ME_m + ME_0C^m = ME_0C^m + M\left(\sum_{j=1}^m M^j E_0 C^{m-j}\right) = \sum_{j=1}^{m+1} M^j E_0 C^{m+1-j}.$$

To bound the extrapolation error,

$$\sum_{m=1}^s E_m = \sum_{m=1}^s \left(\sum_{j=1}^m M^j E_0 C^{m-j}\right) = \sum_{\ell=0}^{s-1}\left(\sum_{j=1}^{s-\ell} M^j\right) E_0 C^\ell = \sum_{\ell=1}^s M^\ell E_0\left(\sum_{i=0}^{s-\ell} C^i\right)$$

Therefore,

$$\|\bar{z}_{k,s} - z^\star\| \le \|z_{k+s} - z^\star\| + \sum_{\ell=1}^s \|M^\ell\| \|E_0\| \|\sum_{i=0}^{s-\ell} C_{(1,1)}^i\|.$$

In the case of $s = +\infty$, we have

$$\|\bar{z}_{k,\infty} - z^\star\| \le \sum_{\ell=1}^\infty \|M^\ell\| \|E_0(\mathrm{Id} - C)^{-1}_{(:,1)}\| = \frac{\|E_0\|}{1 - \sum_i c_i} \sum_{\ell=1}^\infty \|M^\ell\|.$$

The fact that $B_s$ is uniformly bounded in $s$ if $\rho(M) < 1$ and $\rho(C) < 1$ follows because this implies that $\sum_{\ell=1}^\infty \|M^\ell\| < \infty$ thanks to the Gelfand formula, and $\sum_{i=0}^\infty C^i = (\mathrm{Id} - C)^{-1}$ and its $(1,1)^{th}$ entry is precisely $\frac{1}{1 - \sum_i c_i}$.

To control the coefficients fitting error $\epsilon_k$, we follow closely the arguments of [32, Section 6.7], since this amounts to understanding the behaviour of the coefficients $c_k$, which are precisely the MPE coefficients. Recall our assumption that $M$ is diagonalisable, so $M = U^\top \Sigma U$ where $U$ is an orthogonal matrix and $\Sigma$ is a diagonal matrix with the eigenvalues of $M$ as its diagonal. Then, letting $u_k \stackrel{\text{def}}{=} Uv_k$,

$$\epsilon_k = \min_{c \in \mathbb{R}^q} \|\sum_{i=1}^q c_i v_{k-i} - v_k\|$$
$$= \min_{c \in \mathbb{R}^q} \|\sum_{i=1}^q c_i \Sigma^{k-i} u_0 - \Sigma^k u_0\| = \min_{g \in \mathcal{P}_q} \|\Sigma^{k-q} g(\Sigma) u_0\| \le \|u_0\| \min_{g \in \mathcal{P}_q} \max_{z \in \lambda(M)} |z|^{k-q} |g(z)|$$

where $\mathcal{P}_q$ is the set of monic polynomials of degree $q$ and $\lambda(M)$ is the spectrum of $M$. Choosing $g = \prod_{j=1}^q (z - \lambda_j)$, we have $g(\lambda_j) = 0$ for $j = 1, \ldots, q$, so

$$\epsilon_k \le \|u_0\| |\lambda_{q+1}|^{k-q} \max_{\ell > q} \prod_{j=1}^q |\lambda_j - \lambda_\ell|. \tag{F.5}$$

The claim that $\rho(C_k) < 1$ holds since the eigenvalues of $C$ are precisely the roots of the polynomial $Q(z) = z^{k-1} - \sum_{i=1}^{k-1} c_j z^{k-1-i}$, and from [32], if $|\lambda_q| > |\lambda_{q+1}|$, then $Q$ has precisely $q$ roots $r_1, \ldots, r_q$ satisfying $r_j = \lambda_j + \mathcal{O}(|\lambda_{q+1}/\lambda_j|^k)$. So, $|r_j| < 1$ for all $k$ sufficiently large. To prove the non-asymptotic bounds on $\epsilon_k$, first observe that $z_{k+1} - z_k = M(z_k - z_{k-1})$ implies $z_{k+1} - z^\star = M(z_k - z_*)$ and $z_{k+1} - z_k = (M - \mathrm{Id})(z_k - z^\star)$. So, letting $\gamma_i = -c_{k,i}/(1 - \sum_i c_{k,i})$ for $i = 1, \ldots, q$ and $\gamma_0 = 1/(1 - \sum_i c_{k,i})$, we have

$$\frac{1}{1 - \sum_i c_{k,i}}\left(v_k - \sum_{i=1}^q c_{k,i} v_{k-i}\right) = \sum_{i=0}^q \gamma_i v_{k-i} = (M - \mathrm{Id}) \sum_{i=0}^q \gamma_i (z_{k-i-1} - z^\star). \tag{F.6}$$

Now, $y \stackrel{\text{def}}{=} \sum_{i=0}^q \gamma_i z_{k-i-1}$ is precisely the MPE update and norm bounds on this are presented in [32]. For completeness, we reproduce their arguments here: Let $A \stackrel{\text{def}}{=} \mathrm{Id} - M$, by our assumption of $\lambda(M) \subset (-1, 1)$, we have that $A$ is positive definite. Then,

$$\|A^{1/2}(y - z^\star)\|^2 = \langle A(y - z^\star), (y - z^\star)\rangle$$
$$= -\langle \sum_{i=0}^q \gamma_i v_{k-i}, (y - z^\star) + w\rangle$$

where $w = \sum_{j=1}^q a_j v_{k-j}$ with $a \in \mathbb{R}^q$ being arbitrary, since by definition of $\gamma$, $\langle \sum_{i=0}^q \gamma_i v_{k-i}, v_\ell\rangle = 0$ for all $\ell = k - q, \ldots, k - 1$. We can write

$$w = \sum_{j=1}^q a_j (M - \mathrm{Id})(z_{k-j-1} - z^\star) = \sum_{j=1}^q a_j (M - \mathrm{Id}) M^{k-j-1}(z_0 - z^\star) = f(M)(z_0 - z^\star)$$

where $f(z) = z^{k-q-1}(z-1) \sum_{j=1}^{q} a_j z^{q-j}$, and we can write

$$y - z^\star = \sum_{i=0}^{q} \gamma_i M^{k-i-1}(z_0 - z^\star) = g(M)(z_0 - z^\star)$$

where $g(z) = z^{k-q-1} \sum_{i=0}^{q} \gamma_i z^{q-i}$. Therefore, $f(z) + g(z) = z^{k-1-q} h(z)$, where $h$ is a polynomial of degree $q$ such that $h(1) = 1$. Moreover, since the coefficients $a_j$ are arbitrary, $h$ can be considered as an arbitrary element of $\tilde{\mathcal{P}}_q$, the set of all polynomials of degree $q$ such that $h(1) = 1$. Therefore

$$\|A^{-1/2}(y - z^\star)\|^2 \leq \|A^{-1/2}(y - z^\star)\| \min_{h \in \tilde{\mathcal{P}}_q} \|M^n h(M)(z_0 - z^\star)\|$$

$$\leq \|A^{-1/2}(y - z^\star)\| \min_{h \in \tilde{\mathcal{P}}_q} \max_{t \in \lambda(M)} |t^n h(t)| \|z_0 - z^\star\|$$

In particular, combining this with (F.6), we have

$$\frac{\epsilon_k}{|1 - \sum_i c_{k,i}|} \leq \|z_0 - z^\star\| \|(\mathrm{Id} - M)^{1/2}\| \rho(M)^n \min_{h \in \tilde{\mathcal{P}}_q} \max_{t \in \lambda(M)} |h(t)|$$

Finally, in our case where $\lambda(M) = [\alpha, \beta]$ with $1 > \beta > \alpha > -1$, it is well known that $\min_{h \in \tilde{\mathcal{P}}_q} \max_{t \in \lambda(M)} |h(t)|$ has an explicit expression (see, for example, [6] or [32, Section 7.3.1]):

$$\min_{h \in \tilde{\mathcal{P}}_q} \max_{z \in \lambda(M)} |h(z)| \leq \max_{z \in \lambda(M)} |h_*(z)|,$$

where $h_*(z) \stackrel{\mathrm{def}}{=} \frac{T_q\left(\frac{2z - \alpha - \beta}{\beta - \alpha}\right)}{T_q\left(\frac{2 - \alpha - \beta}{\beta - \alpha}\right)}$ where $T_q(x)$ is the $q^{th}$ Chebyshev polynomial and it is well known that

$$\min_{h \in \tilde{\mathcal{P}}_q} \max_{z \in [\alpha, \beta]} |h(z)| \leq 2\left(\frac{\sqrt{\eta} - 1}{\sqrt{\eta} + 1}\right)^q \tag{F.7}$$

where $\eta = \frac{1 - \alpha}{1 - \beta}$. $\qquad\square$