[Reviews · NeurIPS 2019]

Reviewer 1



In this paper the authors analyze the trajectories of variables in ADMM, and make use of these descriptions to propose an adaptation of ADMM. I really liked this paper. The presentation of the paper is ok, in the sense of content and explanations, although there are lots of typos and weird sentences (see some below). In the definition of partial smoothness, you need a function, a point, and a manifold M_x. However in line 92, it reads "Suppose R \in PSF_x(M^R_x)", and the set M^R_x is not defined above (and the same for J) I would have liked to see the trajectories in the experimental section as well, since Fig 1 seems to be for a toy example. Specially for the case not included in the theorems (R and J not polyhedral, but spiral trajectory). Minor comments: Please take a look at this sentence in line 50. "Given an arbitrary [...] converges faster to z*". It's missing a verb. Line 19: "becomes" should be "became", and the use of "owing to" is weird. I would suggest something like "ADDM was first proposed in [13] and has recently gained increasing popularity due to [6]." Line 118: depending the leading eigenvalue -> depending ON the leading eigenvalue Line 125: when it failures -> when it fails Line 128: The above scheme can reformulated -> The above scheme can BE reformulated Line 144: implies -> imply Also take a look at the sentence in line 167: "As E_s,q contains the sum ...". The verb is also missing in this sentence.

Reviewer 2



There are a great many papers written on the ADMM algorithm, and it is often very difficult to see and gauge the novelty of the submission. Thankfully, this paper actually does make a very nice contribution, which is clearly articulated, so I am happy to be able to support this work. The paper is well written and organised. The key observation is highlighted immediately so that the reader can see what the contribution (and therefore value) of the work is. A new algorithm naturally arises, theoretical results are stated to support the observation and algorithm, and there is a good number of numerical experiments to support the paper. I think this paper will be of wide interest to the community.

Reviewer 3



The paper talks about the trajectory of z_k in ADMM under convex setup. Authors prove that, near the local stationary point, v_k = z_k - z_{k-1} equals to a rotation of v_{k-1} plus some smaller order terms. When objective functions are polyhedral, smaller order terms can be dropped off. Moreover, when either of objective functions is smooth and A is full-rank square matrix, the rotation matrix has real eigenvalues for large enough gamma. Under this result, authors show some examples that ADMM with inertial acceleration fails in. In particular, authors study the Lasso problem and show when gamma is small, inertial acceleration does not work out. Motivated from this phenomenon, authors further propose A3DMM to do extrapolation, which is also tested in experiments. This submission has good quality and clarity overall. The analysis is also significant. But I have few concerns: 1, why do we focus on z_k, an intermediate variable of ADMM, instead of psi_k. If similar results can be proved for dual variable, and acceleration step is applied on either x_k (or y_k) or psi_k (which is usually the case), then it makes more sense to me. 2, for the theoretical analysis of z_k, I don't see when direction of z_k will be a straight line. The straight line should correspond to M = I, but this result is not discussed in the paper. 3, for Proposition 2.4, authors claim that if gamma is smaller than threshold, M has complex eigenvalues. but trajectory could still be spiral (or straight line? again, why straight line is not clear). How does this argument predict the behavior of inertial ADMM? I don't see the threshold of gamma distinguishes the behavior of z_k clearly. Also, authors need explain why 2.4 implies straight line for large gamma. 4, It's not clear if the failure of inertial acceleration is due to the z_k-z_{k-1} term from the experiments. Authors should show some iteration trajectories on this term from figures. For many variant ADMM, it's common to have step size parameter before gamma, and analysis will impose conditions on step size. It's not surprising to see inertial ADMM fails for small gamma. To claim the failure is because of direction z_k - z_{k-1}, authors should provide evidence. 5, different from analyzing direction of z_k, it seems that some extra conditions are imposed on M when showing convergence of A3DMM. However, M only has existence argument in previous section. How those conditions on M reduce the function candidates is not very clear. The experiments also ignore the condition on M.

[Author Response · NeurIPS 2019]

We are glad that all the reviewers generally appreciated the significance of our contributions.

**Reviewer #1** We thank the reviewer very much for your positive review and pointing out the typos. Regarding the
trajectory of problems considered in the experiments, we will provide detailed discussion and extra numerics in the
final version of the paper and supplementary material. However, we need to point out that as the problems are in high
dimension, it is impossible to visualize the trajectory and we can only provide the property of $\cos(\theta_k)$ same as the first
row of Figure 2 in the current submission.

**Reviewer #2** We thank the reviewer very much for your positive review and valuable suggestions on discussing previous
works. Indeed, over the past decades, numerous works on ADMM are proposed in the literature, we will rewrite our
discussions and try to include as many related works as possible.

**Reviewer #3** We appreciate the reviewer's overall positive comments on our paper. Below we first explain the reason
why we focus on $z_k$ and then reply to the reviewer's concerns.

It is well-known that ADMM is equivalent to applying Douglas–Rachford (DR) splitting method to the dual form of the
optimization problem at hand. Please find Section C.1 of our supplementary material for detailed discussion.

- For DR to the dual problem, the method can be written as a fixed-point iteration in terms of only $z_k$. Moreover,
such $z_k$ can be expressed by $\psi_k$ and $x_k$ that are generated by ADMM.
- One way to accelerate ADMM is via its equivalence with DR, as DR is further equivalent to Proximal Point
Algorithm (PPA) whose inertial version is well studied in the literature. Given inertial PPA, one can easily derive
the inertial version of DR, hence inertial ADMM which is exactly the scheme discussed in Section 3.

Owing to the above reasons, we opted to use $z_k$ to discuss the trajectory of ADMM. Replies to the reviewer's concerns:

1. Thanks for pointing this out. Our accelerate scheme can also be applied to the dual variable $\psi_k$ and primal variables
$x_k, y_k$. The advantage of considering $z_k$ is that you only need to store the past of $z_k$ and extrapolate $z_k$, since the
update of $y_k$ in Algorithm 1 depends only on $z_k$. If we consider applying the proposed scheme to the standard
ADMM, i.e. Eq. (1) of the submission, we need to store the past points of $\psi_k$ and $y_k$ and extrapolate both of them,
since the update of $x_k$ depends on $\psi_k$ and $y_k$. Focusing on $z_k$ is simpler to implement in practice. We will add a
remark on this in the final version of the paper.
2. The (eventual) trajectory of $z_k$ depends on the leading eigenvalue of $M$: the local trajectory is a straight line when
the leading eigenvalue of $M$ is real – this is due to the analysis of [Sections C.2 and C.3] in the supplementary, we
will add a few lines to clarify. The case of complex leading eigenvalue is more delicate: In general, one can observe
that this trajectory is a spiral (see the example of Section 3), however, theoretical characterisations of the trajectory
will depend on specific properties of the functions $R$ and $J$ and hence, we only gave a characterisation when both
terms are polyhedral. The point we wanted to make is that typical intuition of inertial has been to extrapolate in the
current direction, but this is not always a good idea because the current direction may be pointing away from the
optimum (if the trajectory is not straight).
3. As said above, when the leading eigenvalue is complex, the trajectory is spiral. Since $\gamma$ is the solo parameter of
ADMM, its choice affects the property of the leading value of $M$, hence determines the trajectory of $z_k$. As a result,
the performance of inertial ADMM discussed in Section 3.
4. Given the inertial scheme considered in the submission, the failure of inertial is due to the trajectory of $z_k$. We will
provide evidence on the angle $\theta_k$ to support our statement. We are not sure what does the step-size here means, as if
this step-size is added in front of all $\gamma$'s in ADMM iteration, then it is just a scaling factor and does not change our
conclusion. To further emphasize the importance of sequence trajectory, we can also write inertial ADMM as a
fixed point iteration and look at its local linearisation matrix $M$, it is then possible to see that if the original ADMM
linearisation has real leading eigenvalue, then $\rho(M) < 1$, while complex leading eigenvalue can cause $\rho(M) > 1$.
We can add this short argument to further reinforce our point.
5. We have to emphasize that the global convergence of our proposed acceleration scheme **does not** need any
assumption on $M$ (see Proposition 4.2), since $M$ can be obtained only locally. Our assumptions on $M$ is only
for establishing local acceleration guarantees, and these assumptions are standard in the literature of minimal
polynomial extrapolation in numerical analysis.

For the three points raised in "Improvements: What would the authors have to do for you to increase your score?"

1. Please see the response to points 2 and 3 above.
2. Please see the response to point 4.
3. We do believe that the comparison against SVRG-ADMM would be unfair since it is a stochastic method, and
our focus is in the deterministic realm. If the "PRSM" refers to Peaceman–Rachford splitting method, we would
like to point out that: First, PRSM is not guaranteed to be better than ADMM. Second, we are currently working
on extending the proposed adaptive acceleration to more general first-order methods, so our adaptive acceleration
can also be applied to PRSM based on our latest result, but we feel that comparisons to other first order methods
is beyond the scope of this current work whose focus is ADMM. Finally, to our knowledge, previous forms of
accelerated ADMM such as "ADMM and Accelerated ADMM as Continuous Dynamical Systems by Franca et al"
target the case where both functions are strongly convex, whereas one of our goals of this work is to move beyond
this case and consider the most general setting, that is both functions can be non-smooth and only convex.

[Meta-Review · NeurIPS 2019]

The authors study the trajectories of intermediate variables in ADMM, and use this to propose a refinement to inertial ADMM. The paper is full of interesting ideas and the reviews were positive. One reviewer expressed concerns about the paper's readability and presentation; I would like the authors to address this to the extent they can. [This meta-review was reviewed and revised by the Program Chairs]